# The long non-coding RNA NEAT1 is a ΔNp63 target gene modulating epidermal differentiation

Claudia Fierro[1,6,7], Veronica Gatti[2,7], Veronica La Banca[1], Sara De Domenico[1], Stefano Scalera [3], Giacomo Corleone[3], Maurizio Fanciulli[3], Francesca De Nicola [3], Alessandro Mauriello [1], Manuela Montanaro[1], George A. Calin [4,5], Gerry Melino [1] & Angelo Peschiaroli [2] ✉

The transcription factor ΔNp63 regulates epithelial stem cell function and maintains the integrity of stratified epithelial tissues by acting as transcriptional repressor or activator towards a distinct subset of protein-coding genes and microRNAs. However, our knowledge of the functional link between ΔNp63 transcriptional activity and long non-coding RNAs (lncRNAs) expression is quite limited. Here, we show that in proliferating human keratinocytes ΔNp63 represses the expression of the lncRNA NEAT1 by recruiting the histone deacetylase HDAC1 to the proximal promoter of NEAT1 genomic locus. Upon induction of differentiation, ΔNp63 down-regulation is associated by a marked increase of NEAT1 RNA levels, resulting in an increased assembly of paraspeckles foci both in vitro and in human skin tissues. RNA-seq analysis associated with global DNA binding profile (ChIRP-seq) revealed that NEAT1 associates with the promoter of key epithelial transcription factors sustaining their expression during epidermal differentiation. These molecular events might explain the inability of NEAT1-depleted keratinocytes to undergo the proper formation of epidermal layers. Collectively, these data uncover the lncRNA NEAT1 as an additional player of the intricate network orchestrating epidermal morphogenesis.

A master regulator of epidermis development is the transcription factor p63[1]. p63 is expressed as multiple isoforms with specific properties, including a full length and an amino-deleted isoform, named TAp63α and ΔNp63α (hereinafter referred as TAp63 and ΔNp63), respectively. The shorter isoform ΔNp63 is highly expressed in the proliferative compartments of glandular, single, and stratified epithelia[2]. In the epidermis, ΔNp63 expression is restricted to the basal compartment and disappears in the suprabasal layers in concomitance with the formation of the cornified envelope[3–5]. The critical role of ΔNp63 in skin development was revealed by the analysis of ΔNp63 knockout mice which, in addition to defects in mammary glands and craniofacial region, display severe defects in the epidermis formation[6].

[1]Department of Experimental Medicine, Tor Vergata Oncoscience Research (TOR), University of Rome "Tor Vergata", Via Montpellier 1, 00133 Rome, Italy. [2]Institute of Translational Pharmacology (IFT), CNR, Via Fosso del Cavaliere 100, 00133 Rome, Italy. [3]UOSD SAFU, Department of Research, Advanced Diagnostics, and Technological Innovation, Translational Research Area, IRCCS Regina Elena National Cancer Institute, Rome, Italy. [4]Department of Translational Molecular Pathology, The University of Texas MD Anderson Cancer Center, Houston, TX 77030, USA. [5]Center for RNA Interference and Non-Coding RNAs, The University of Texas MD Anderson Cancer Center, Houston, TX 77030, USA. [6]Present address: Translational Pediatrics and Clinical Genetics Research Division, Bambino Gesù Children's Hospital, IRCSS, Piazza Sant'Onofrio, 4, Rome, Italy. [7]These authors contributed equally: Claudia Fierro, Veronica Gatti. ✉e-mail: angelo.peschiaroli@cnr.it

At a functional level, omics approaches have allowed the identification and characterization of ΔNp63-dependent transcriptional signature, which includes protein-coding genes involved in regulating cell adhesion, cell metabolism, cell–matrix interactions, senescence, stem cell function, and activating the early step of the differentiation program[5,7–12]. At a molecular level, ΔNp63 directly regulates the expression of many genes, acting as transcriptional activator or repressor by interacting with distinct epigenetic modulators and chromatin remodeling factors[11,13]. For instance, ΔNp63-DNMT3a complex maintains high levels of DNA hydroxymethylation at the enhancers of epidermal genes[14]. ΔNp63 can also interact with chromatin modifiers enzymes associated with non-permissive transcription state, such as the histone deacetylases HDAC1/HDAC2 and the epigenetic modifier ACTL6a, acting thus as a transcriptional repressor[15,16].

During the last decade, lncRNAs have emerged as crucial regulators of a variety of physiological processes, including somatic lineage specification and differentiation[17,18]. So far, only few lncRNAs have been described to play a role in controlling epidermal differentiation. Examples of such lncRNAs are the anti-differentiation ncRNA ANCR, the RNA LINC00941 and SMRT-2, whose expression is necessary for maintaining skin homeostasis, even though their mode of action remains unclear[19,20]. Although initial studies identified TINCR as terminal differentiation-induced ncRNA required for skin differentiation, recent evidence indicate that TINCR RNA encodes for an ubiquitin-like protein and therefore its requirement for skin differentiation likely relies on its ability to act as protein coding gene[21].

Although numerous ΔNp63 target genes (protein-coding genes and microRNAs) have been characterized so far, how and whether ΔNp63 regulates long non-coding RNA (lncRNAs) expression is poorly known. Here, starting from a custom-designed lncRNAs microarray[22], we identified the lncRNAs NEAT1 and MALAT1 as a bona fide ΔNp63 transcriptional targets. The molecular details of ΔNp63/NEAT1-MALAT1 axis as well as its impact on epidermis differentiation are presented herein.

## Results

### ΔNp63 represses NEAT1 and MALAT1 expression

To test whether ΔNp63 regulates lncRNAs expression, we performed a lncRNAs custom-designed microarray assay utilizing RNA extracted from primary and tumors cells upon ΔNp63 silencing (Fig. 1A and S1A). We utilized FaDu and A253 Squamous Cell Carcinoma (SCC) cell lines, HCC1954 basal breast carcinoma cell line, and two human primary cells, keratinocytes (HEKn) and mammary epithelial cells (HMEC). All these cell types express exclusively the p63 isoform ΔNp63 (Fig. S1A)[23]. We identified a significant number of annotated lncRNAs that were downregulated or upregulated in a cell type specific manner upon p63 depletion (|log2 fold change (FC)| > 1, adjusted $p$ value < 0.05) (Fig. 1B). As shown in Fig. S1B and Table S1, we identified lncRNAs whose expression is downregulated (MIR17HG, AY827612 and BCRNY) or upregulated (MALAT1 and FTX) in all cell lines tested. We reasoned that those lncRNAs might represent bona-fide ΔNp63 transcriptional targets. We decided to prioritize our investigation on MALAT1 (encoding metastasis-associated lung adenocarcinoma transcript 1, also known as NEAT2) since its modulation upon p63 silencing in primary cells might be indicative of an unexpected function of this lncRNA in a non-pathological context. Furthermore, we decided to include in our investigation the lncRNA NEAT1 (Nuclear Enriched Abundant Transcript 1, whose probe was not included in the lncRNAs custom-designed microarray), since several observations suggest that MALAT1 and NEAT1 might be functionally interconnected. In detail, these observations are: (i) the analysis of the 3D chromatin interactions in human cells which localizes MALAT1 genomic locus in close proximity to the NEAT1 locus[24] (ii) the adjacent nuclear localization of speckles (MALAT1 positive foci) and paraspeckles (NEAT1 positive foci)[25]; (iii) the ability of NEAT1 and MALAT1 to co-regulate a subset of common genes[26].

We firstly validated the upregulation of NEAT1 RNA levels in primary and tumors cells upon p63 silencing (Fig. S1C). Then, we tested the impact of ΔNp63 on NEAT1 RNA modulation. We transfected primary keratinocytes with two different siRNA oligos targeting ΔNp63 mRNA. As shown in Fig. 1C, the specific depletion of ΔNp63 isoform markedly increased MALAT1 and NEAT1 RNA levels in primary keratinocytes. NEAT1 gene encodes two isoforms: a short isoform of approximately 3,7 kb and a long one of ~21,7 kb (NEAT1_2). This long isoform is transcribed from the same promoter and is required for the formation of the paraspeckles in vitro and in vivo[27–30]. By utilizing primers which specifically detect NEAT1 long isoform, we confirmed that NEAT1_2 expression undergoes similar modulation upon ΔNp63 silencing (Fig. 1D).

### ΔNp63 exploits an HDAC-dependent mechanism to repress NEAT1 and MALAT1 expression

To determine whether the modulation of NEAT1 and MALAT1 expression upon ΔNp63 silencing is the result of a direct binding of ΔNp63 to their promoters, we searched for p63-binding sites within 10 kb upstream and downstream of NEAT1 and MALAT1 transcription start sites (TSS) in publicly available p63 Chip-Seq dataset of primary keratinocytes (GSM1446927). By Chip-seq dataset analysis (Fig. 1E) and confirmatory ChIP assay (Fig. 1F and S2A) we found that ΔNp63 is able to physically occupy a canonical p63 DNA binding site located in the proximal promoter of NEAT1 and MALAT1 genes (Fig. S2B). To repress the expression of its target genes ΔNp63 exploits various epigenetic-based mechanisms, including the recruitment of the histone deacetylases HDAC1/HDAC2[16]. To test whether HDAC activity might be involved in the ΔNp63-mediated repression of NEAT1 and MALAT1, we treated primary keratinocytes with Givinostat, a specific and potent inhibitor of histone deacetylases (HDAC) activity. As shown in Fig. S2C, we observed a dose-dependent increase of NEAT1 and MALAT1 RNA levels upon Givinostat treatment. To further investigate the involvement of HDAC in the ΔNp63-NEAT1/MALAT1 axis, we performed a ChIP assay utilizing the HDAC1 specific antibody and found that HDAC1 is recruited to the p63 DNA binding region located in the NEAT1 and MALAT1 promoters (Fig. 1G and S2D). Remarkably, p63 silencing decreases HDAC1 occupancy in NEAT1 and MALAT1 promoters (Fig. S2E) and induces a marked increase of histone H3 acetylation on NEAT1 and MALAT1 promoter regions (Fig. 1H). The analysis of publicly available H3K27 Chip-Seq dataset of primary keratinocytes in proliferating and differentiating conditions indicates that during epidermal differentiation the decrease of ΔNp63 expression (see Fig. 2A) is parallel with the upregulation of transcription permissive mark H3K27ac (Fig. S2F). These data clearly indicate the ability of ΔNp63 to modulate NEAT1 and MALAT1 promoters acetylation which ultimately lead to their non-permissive transcription state.

### NEAT1 and MALAT1 expression is modulated during epidermal differentiation

ΔNp63 levels are tightly regulated during skin differentiation, being its levels are high in proliferating keratinocytes and decreasing in concomitance with the activation of epidermal differentiation program[5]. To test whether NEAT1 and MALAT1 expression is inversely correlated with that of ΔNp63 in a physiological context, we analyze their expression during keratinocyte differentiation utilizing the well-established model of primary keratinocytes that undergo calcium-induced differentiation in vitro. As shown in Fig. 2A, B, the activation of the epidermal differentiation program markedly increased MALAT1 and NEAT1 RNA levels, in concomitance with the downmodulation of ΔNp63 and the upregulation of the differentiation marker keratin 10 (K10). NEAT1_2 RNA levels show the same behavior of total NEAT1, as revealed by the RT-qPCR analysis (Fig. 2B) and RNA-seq data (Fig. 2C). Since the assembly of paraspeckles is strictly dependent on NEAT1_2, we asked whether paraspeckles assembly could be modulated during

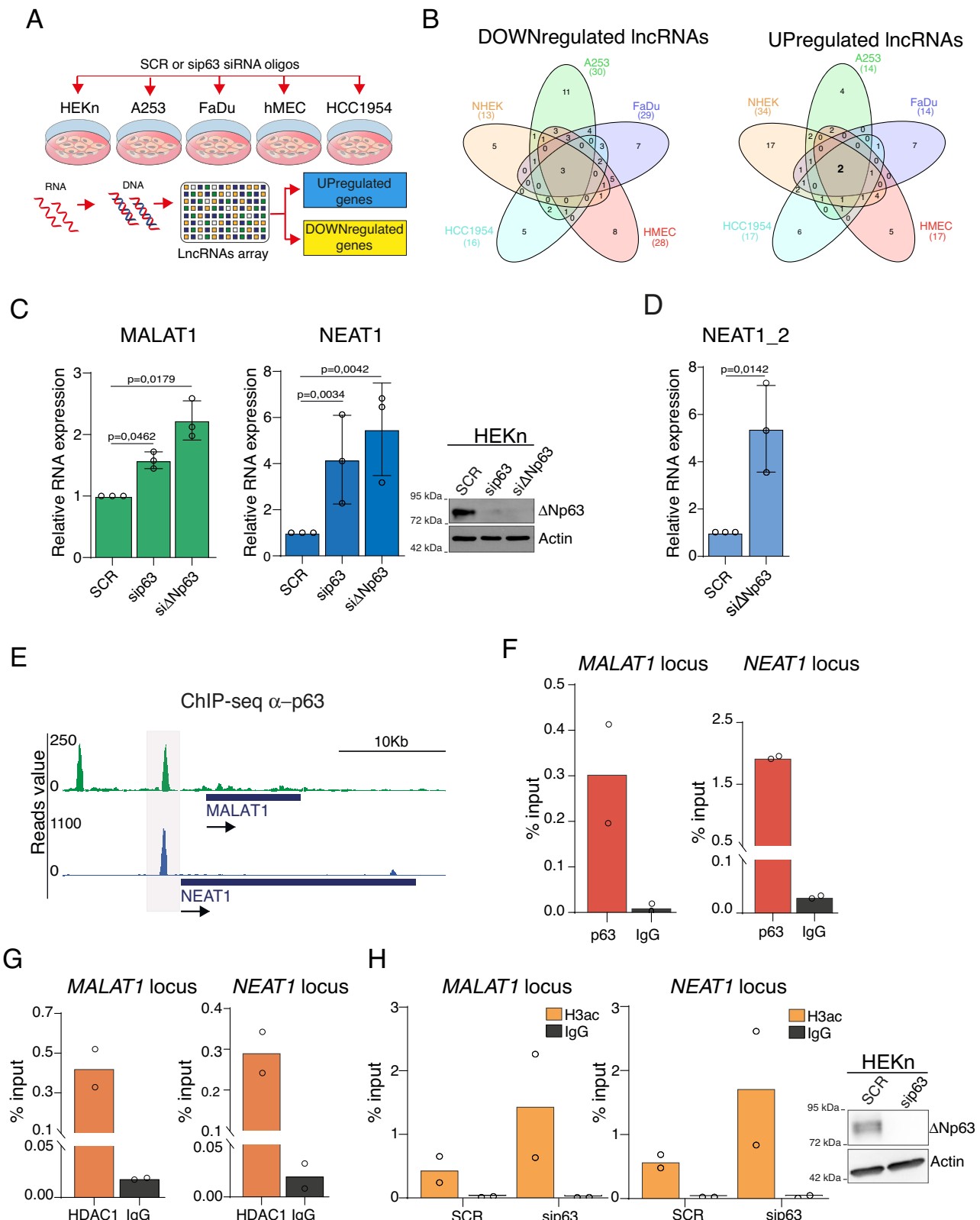

keratinocytes differentiation. To this aim, we performed RNA–fluorescence in situ hybridization (RNA-FISH) in proliferating or differentiated keratinocytes. We observed a marked increase in the NEAT1 RNA–FISH signal, which was distributed in a characteristic punctate pattern in HEKn cells after calcium-induced differentiation (Fig. 2D). By ultraresolution confocal analysis we found that FISH probe detecting the 5' common region of NEAT1_1 and NEAT1_2 is exclusively detected in foci overlapping with those detected by NEAT1_2 specific probe (Fig. S3A) suggested that most of the NEAT1 signal detected in differentiated keratinocytes is associated with the NEAT1_2 isoform. In line with this evidence, RT-qPCR analysis revealed that differentiated keratinocytes mostly express NEAT1_2 isoform (Fig. S3B). Notably, NEAT1 signal co-localized with the paraspeckles marker SPFQ, which re localizes into NEAT1-positive foci in

**Fig. 1 | ΔNp63 represses NEAT1 and MALAT1 lncRNAs expression by an HDAC-dependent mechanism. A** Schematic illustration of the lncRNAs array-based approach utilized to identify lncRNAs regulated by p63. Briefly, the indicated primary (HEKn and hMEC) and cancer cell types (A253, FaDu, HCC1954) were transfected with scramble (SCR) or siRNA oligos targeting p63 mRNA (sip63), and the cDNA utilized for hybridization assay of a custom-made lncRNAs array. **B** Venn diagrams showing shared downregulated and upregulated lncRNAs in sip63-transfected cells. **C** Human primary keratinocytes (HEKn) were transfected with siRNA targeting p63 (sip63), ΔNp63 isoform (siΔNp63) or non-relevant mRNA (SCR). MALAT1 and NEAT1 RNA levels were quantified by RT-qPCR (left panel). Data shown are the mean of three ($n = 3$) independent biological replicates ± SD. $p$ value was calculated using two-tailed unpaired Student's $t$ test. In parallel, protein lysates from transfected cells were analyzed by western blotting using antibodies to the indicated proteins (right panel). **D** HEKn cells were transfected with siRNA targeting ΔNp63 isoform (siΔNp63) or non-relevant mRNA (SCR). NEAT1 long isoform (NEAT1_2) RNA levels were quantified by RT-qPCR. Data shown are the mean of three ($n = 3$) independent biological replicates ± SD. $p$ value was calculated using two-tailed unpaired Student's $t$ test. **E** ChIP-seq enrichment of endogenous p63 at MALAT1 and NEAT1 genomic loci in HEKn cells (GSM1446927). **F** ChIP-qPCR showing ΔNp63 occupancy at the p63 binding site of MALAT1 and NEAT1 genomic loci. Average values from $n = 2$ biological replicates measured using three technical replicates are plotted. **G** ChIP-qPCR showing endogenous HDAC1 occupancy at MALAT1 and NEAT1 genomic loci in HEKn cells. Average values from $n = 2$ biological replicates measured using three technical replicates are plotted. **H** ChIP-qPCR showing Histone H3 acetylated (H3ac) occupancy at MALAT1 and NEAT1 genomic loci in HEKn cells transfected with scramble (SCR) or siRNA oligo targeting p63 (sip63) (left panels). Average values from $n = 2$ biological replicates measured using three technical replicates are plotted. In parallel, protein lysates from transfected cells were analyzed by western blotting using antibodies to the indicated proteins (right panel). Source data are provided as a Source Data file.

differentiated keratinocytes (Figs. 2D, E). RNA-FISH also confirmed the upregulation of MALAT1 in differentiated keratinocytes (Fig. S3C). To confirm these data in human tissues we performed RNA-FISH and immunofluorescence (IF) analysis in human skin samples. As shown in Fig. 2F, ΔNp63 expression is mainly restricted to the basal compartment and decreased in the upper layer of human epidermis, while NEAT1 and MALAT1 RNA signals are exclusively detected in the suprabasal ΔNp63-negative cells. Based on this data, we concluded that during the onset of skin differentiation ΔNp63 downmodulation is associated with the upregulation of NEAT1 and MALAT1 RNA levels and the concomitant assembly of NEAT1-associated paraspeckles.

## NEAT1 depletion affects the expression of epidermal differentiation genes

Several evidence indicates that NEAT1 and MALAT1 might affect gene expression and several mechanisms underlying this function have been proposed[31,32]. To characterize the impact of NEAT1 and MALAT1 on the expression profile of human keratinocytes upon differentiation we performed RNA-seq analysis in differentiated keratinocytes upon depletion of NEAT1 or MALAT1 by LNA-GapmeR-mediated approach. We confirmed NEAT1 and MALAT1 depletion by RT-qPCR analyses (Fig. S4A). Furthermore, NEAT1 depletion leads to paraspeckles disassembly in differentiated keratinocytes, confirming the importance of NEAT1 as essential component of paraspeckles (Fig. S4B). Parallel to paraspeckles disgregation, NEAT1 depletion resulted in significant expression level changes in ~370 genes (adjusted $p$ value < 0.05) with 219 genes downregulated and 152 genes upregulated (Fig. 3A, B). Conversely, MALAT1 depletion affects the expression of a smaller number of genes (~100) which show no overlapping with the list of genes affected by NEAT1 depletion (Fig. 3B). While we did not find any gene enrichment for GO terms in LNA-MALAT1 deregulated genes, GO term analysis of the NEAT1-dependent transcriptional profile revealed the functional link between NEAT1 expression and key cellular processes regulating epidermal homeostasis (Fig. 3C and S5A). In detail, LNA-NEAT1 downregulated genes were enriched for GO terms associated with epidermal development, cornification process, keratinocytes differentiation and lipid metabolic process (Fig. 3C, D).

Gene set enrichment analysis (GSEA) comparing the LNA-NEAT1 downregulated gene set to a calcium-differentiated keratinocyte gene set (GSE18590) confirmed the enrichment and correlation between NEAT1 expression and genes induced in calcium-differentiated keratinocytes ($p$ value = 0,0) (Fig. 3E). Remarkably, LNA-NEAT1 downregulated gene set is enriched for GO terms associated with several skin diseases characterized by alterations of the differentiation program, such as Ichthyosis (Vulgaris and Netherton syndrome), SCC or Pachyonychia congenita (Fig. 3F).

To further place NEAT1-dependent transcriptome in the context of skin differentiation, we compared LNA-NEAT1 deregulated genes to several published gene sets of known regulators of epidermal differentiation. We found that NEAT1 profile is significantly enriched to genes whose expression is downmodulated upon silencing of key player of skin differentiation, such as TP63, ZNF750 and KLF4 (Fig. 3G). Importantly, NEAT1 gene profile did not correlate with genes upregulated upon silencing of these epidermal regulators (Fig. S5B). To further confirm these data, we analyzed the entire NEAT1 gene set with knowledge-based IPA Upstream Regulator analysis. This analysis unveiled KLF4 as the top upstream transcriptional regulator that can explain the observed gene expression changes in our RNA-seq (Fig. 3H). By an additional gene regulatory network tool (DoRothEA), we confirmed the correlation between NEAT1-downregulated genes and target genes of epidermal transcription factors involved in skin differentiation and epidermal (Figs. S5C, S5D). Collectively, these data indicate that NEAT1 depletion impacts the expression of key regulators of skin differentiation.

## NEAT1 trans genomic profile in differentiated keratinocytes

Several data indicated that NEAT1 localizes to the DNA regions of highly expressed genes[26,33], suggesting that NEAT1 genomic binding and transcription activation might be interrelated. Based on this consideration, we decided to profile the trans genomic binding sites of endogenous NEAT1 upon epidermal differentiation, a biological process characterized by an extensive and substantial transcriptional induction of epidermal differentiation genes. To this aim, we adopted the chromatin isolation by RNA purification (ChIRP) assay[34]. As schematically illustrated in Fig. 4A this procedure uses biotinylated capture oligonucleotides capable to hybridize to NEAT1 to isolate RNA-associated DNA which is then analyzed by NGS. First and foremost, we checked the reliability of our ChIRP data by analyzing the ability of NEAT1 to localize into *NEAT1* and SP3 genes, as these genomic loci have been previously identified as NEAT1 DNA trans binding loci[26]. As shown in Figs. S6A, S6B, we confirmed NEAT1 localization on these genomic regions, and more importantly, this ChIRP-based DNA enrichment is NEAT1 specific, as the use of LacZ oligos does not produce any significant signal. Although previous report observed an enrichment of NEAT1 signals on *MALAT1* locus[26], we did not observe such localization in differentiated keratinocytes (Fig. S6A). At a global level, we found that NEAT1 trans genomic binding sites on DNA are preferentially localized in the gene bodies (Fig. S7A) with a high percentage of peaks localized in the introns, likely reflecting the role of NEAT1 to modulate RNA splicing. As previously described in a different cellular context[26], we found that NEAT1 localizes to transcriptional start sites (TSSs) and to a lesser extent to the transcriptional termination sites (TTSs) (Fig. 4B). By comparing the NEAT1-associated DNA regions encompassing the TSSs to previously reported datasets reporting covalent histone marks, we found a significant enrichment of NEAT1 TSSs targets for the active chromatin-associated histone H3 modification H3K4me3 (3523 out of 6464) (Fig. 4C). Conversely, H3K27me3, a histone mark for silent chromatin, is poorly enriched (623 out of 6464 NEAT1 TSSs targets) (Fig. 4C).

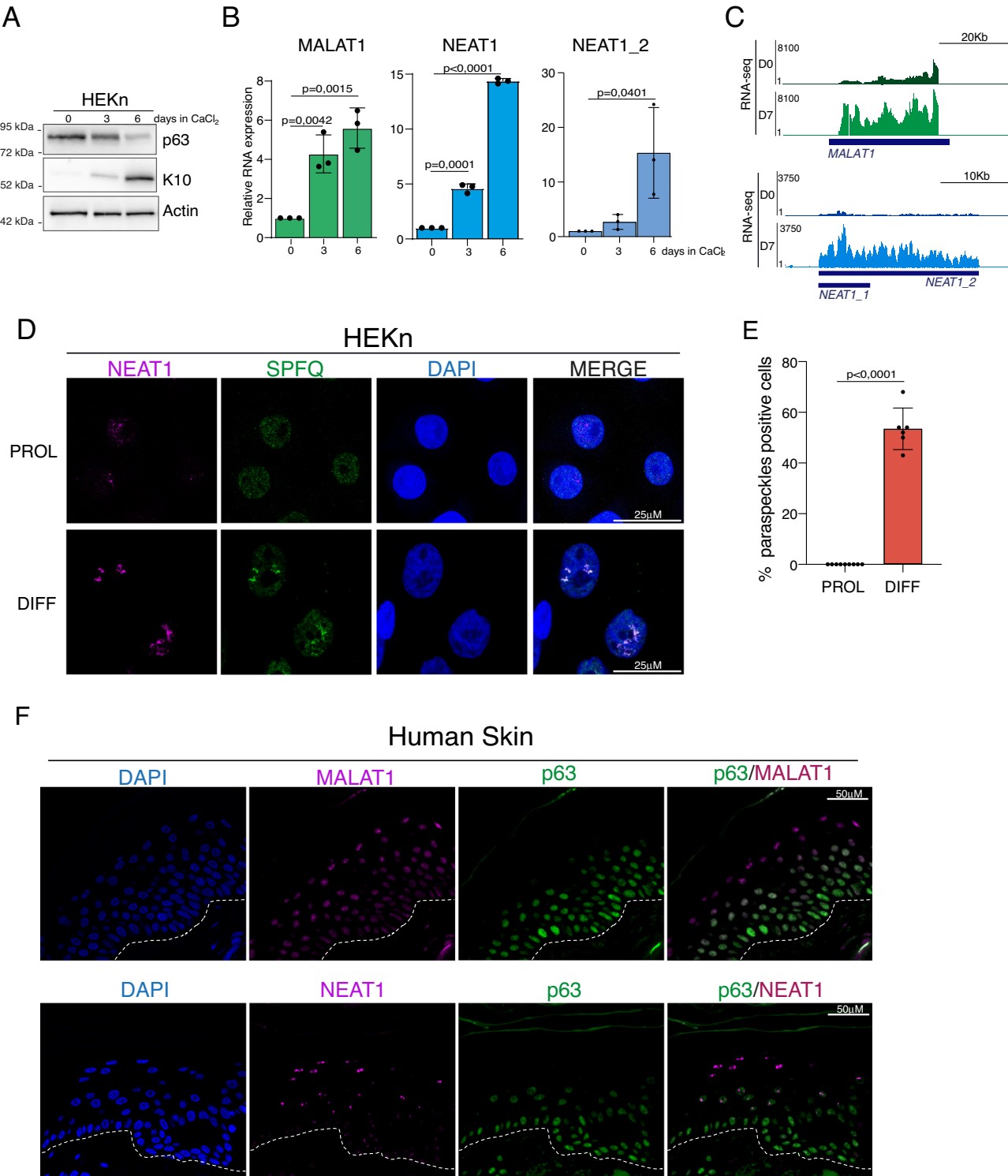

**Fig. 2 | NEAT1 and MALAT1 expression is induced during epidermal differentiation.** Analysis of cell lysates (**A**) or total RNA (**B**) extracted from human primary keratinocytes (HEKn) at different time points (0, 3, 6 days) upon CaCl$_2$ treatment. The protein lysates were immunoblotted for the indicated proteins. MALAT1, NEAT1 and NEAT1_2 RNA levels were quantified by RT-qPCR. Data shown are the mean of three (*n* = 3) independent biological replicates ± SD. *p* value was calculated using two-tailed unpaired Student's *t* test. **C** MALAT1 and NEAT1 RNA-seq reads in proliferating keratinocytes (D0) or upon 7 days of CaCl$_2$ treatment (D7) (GSM1446880, GSM1446883). **D** RNA FISH of endogenous NEAT1 (magenta) and immunostaining of SFPQ (green) in proliferating (PROL) or differentiated (DIFF)

HEKn cells. Nuclei were visualized by DAPI (blue) counterstaining. Scale bars, 25 µm. **E** Quantification of paraspeckle structures. Data shown are the mean of 170 nuclei (*n* = 170) analysis ± SD. *p* value was calculated using two-tailed unpaired Student's *t* test. The experiment was repeated twice with similar results (*n* = 2). **F** Representative confocal images of MALAT1 and NEAT1 localization by RNA FISH in human skin tissue. Basal cells and nuclei were revealed by p63 immunostaining and DAPI counterstaining, respectively. Dotted white line demarcates the epidermal basement membrane. Scale bars, 50 µm. The experiment was repeated twice with similar results (*n* = 2). Source data are provided as a Source Data file.

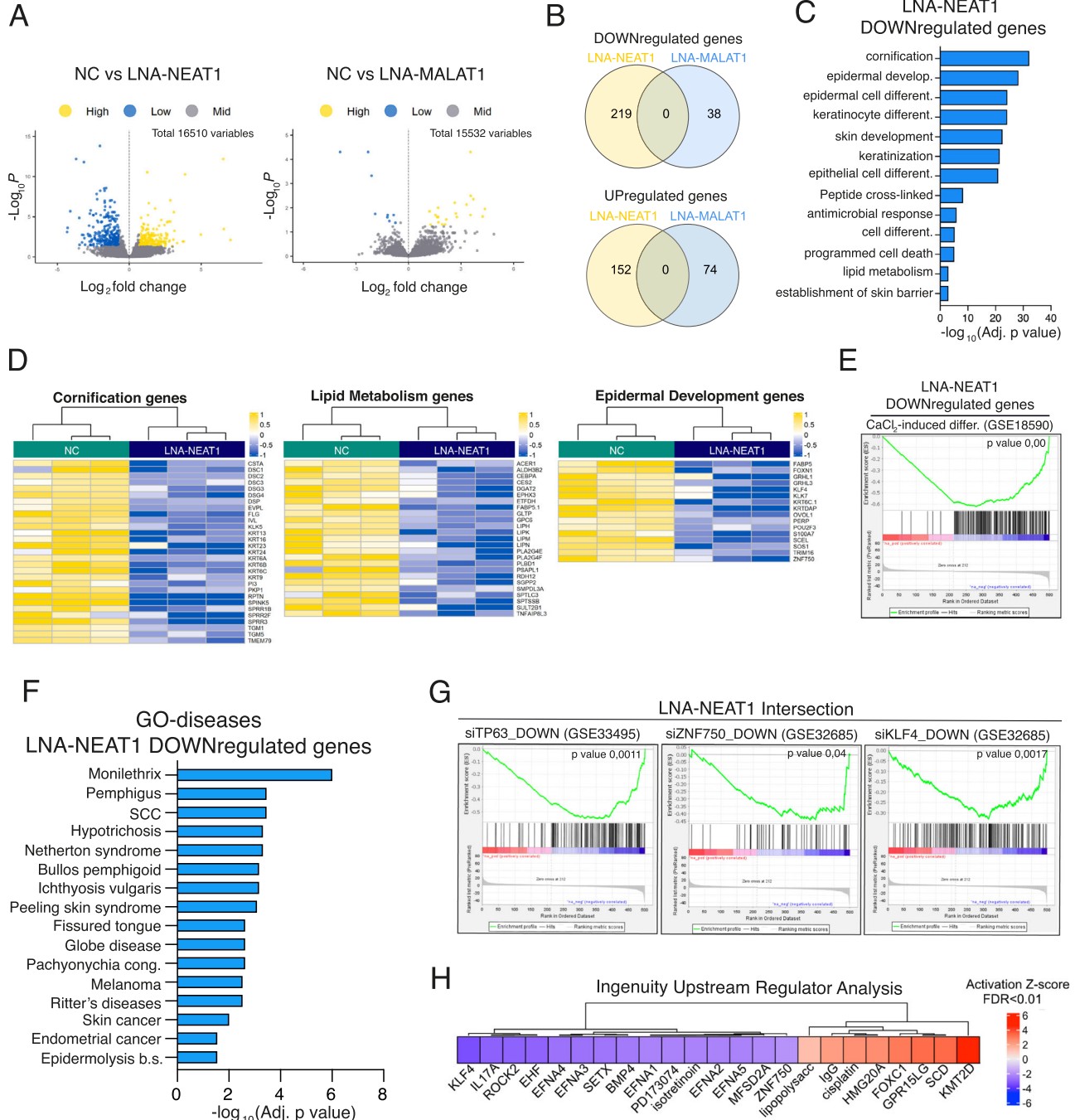

**Fig. 3 | NEAT1 controls the expression of key epidermal differentiation genes.**
**A** Volcano plots displaying gene expression Log2 fold changes and their respective statistical score (*p* value) of negative LNA GapmeR control (NC) vs LNA GapmeR targeting MALAT1 (LNA-MALAT1) or NEAT1 (LNA-NEAT1) transfected differentiated HEKn cells. Differential expression analysis was carried out using DESeq2 package. Volcano plot was generated with Enhanced Volcano R package and statistical analyses were carried out using R software. *P* = FDR (False Discovery Rate) adjusted *p* value. **B** Venn diagram showing the shared downregulated genes and upregulated genes in LNA-NEAT1 or LNA-MALAT1 transfected HEKn cells. **C** Barplot showing the top gene ontology (GO) terms for Biological Process of the downregulated genes upon LNA-NEAT1 transfection of differentiated HEKn cells. GO terms were ordered by FDR (False Discovery Rate) adjusted *p* value calculated by ShinyGO version 0.76.

**D** Heatmap showing the unsupervised hierarchical clustering of genes associated to the top GO terms in negative control (NC, green) or LNA-NEAT1 (violet) transfected differentiated HEKn cells. Color scheme: yellow (highest) blue (lowest) VSD score. **E** GSEA of genes downregulated in NEAT1 depleted keratinocytes against a calcium-differentiated keratinocyte gene set (GSE18590). **F** Bar plot showing the top GO DISEASE terms of genes downregulated in NEAT1 depleted keratinocytes. GO terms were ordered by FDR (False Discovery Rate) adjusted *p* value calculated by ShinyGO version 0.76. **G** GSEA of LNA-NEAT1 downregulated genes set against three keratinocytes differentiation signatures resulted by p63 (GSE33495), ZNF750 (GSE32685) or KLF4 (GSE32685) depletion. **H** IPA Upstream regulator analysis of NEAT1 profile RNA.

## NEAT1 binds to the promoters of key epidermal differentiation genes, favoring their efficient expression

To place NEAT1 genomic binding sites in the context of skin differentiation, we performed an intersection analysis between the NEAT1 bound genes and the RNA profile of differentiated keratinocytes (GSM1446880 vs GSM1446883). We found that NEAT1 binds the promoter of 651 and 116 genes whose expression is induced and repressed during keratinocytes differentiation, respectively. GO

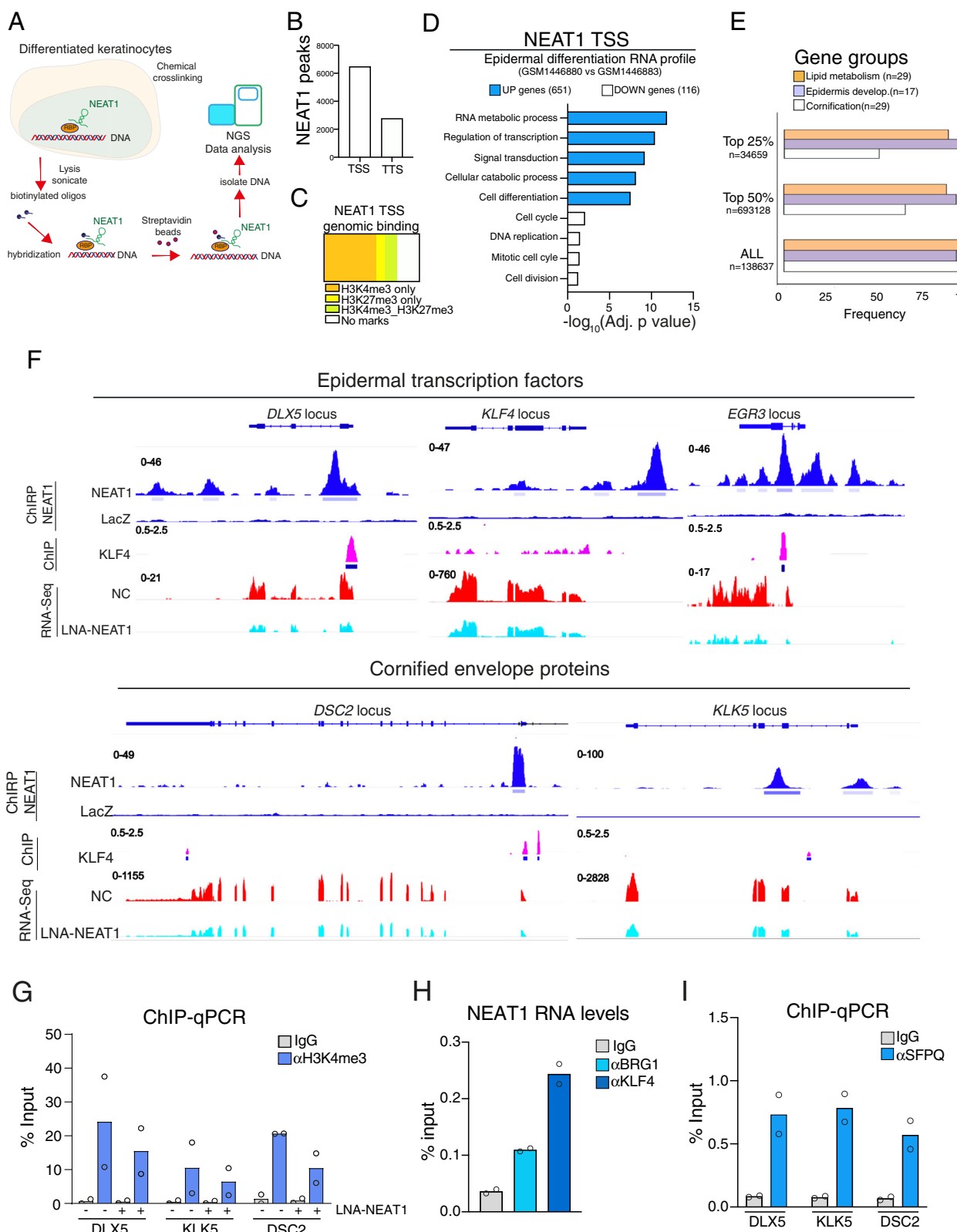

term analyses revealed that the 651 genes significantly categorize in GO terms associated with regulation of transcription and cell differentiation (average FDR enrichment 10e⁻¹⁰) (Fig. 4D). Along the same line, the subgroup of genes bound and activated by NEAT1 categorizes in GO terms associated with epidermis development, establishment of skin barrier and skin development (Figs. S7B, S7C). Furthermore, NEAT1 activated epidermal genes (*e.g.* cornification,

epidermal development and lipid metabolism genes, see Fig. 3D) are enriched in top 25% of NEAT1 trans genomic sites (Fig. 4E). These NEAT1 epidermal genes include genes critically involved in epidermal differentiation and integrity, such as the epidermal transcription factors (ZNF750, KLF4, DLX5, EGR3), the serine proteases (KLK5, KLK6 and KLK7) and the component of desmosome DSC2 (Figs. 4F, S7D, S7E). Remarkably, NEAT1 binding on these epidermal genes

**Fig. 4 | NEAT1 binds to the promoters of key epidermal differentiation genes.**
**A** Schematic illustration of NEAT1 ChIRP assay performed in differentiated HEKn cells (four days of $CaCl_2$ treatment). **B** NEAT1 ChIRP enrichment over Transcription Starting Site (TSS) and Transcription Termination Sites (TTS) in differentiated keratinocytes. **C** Intersection of NEAT1 ChIRP enrichment over TSS against the indicated covalent histone modifications signatures in differentiated keratinocytes (H3K4me3, active genes, GSE98483; H3K27me3, transcriptionally silent genes, GSE175068). **D** Barplot showing the top gene ontology (GO) terms for Biological Process of the intersection analysis between NEAT1 binding sites over TSSs and the RNA profile of differentiated keratinocytes (GSM1446880). GO terms were ordered by FDR (False Discovery Rate) adjusted *p* value calculated by ShinyGO version 0.76. **E** Gene enrichment analysis of the indicated epidermal genes in the entire (ALL), top 50% and top 25% NEAT1 binding sites subgroups. **F** RNA-seq reads in control (NC) and NEAT1 depleted keratinocytes (LNA-NEAT1) together with NEAT1 ChIRP enrichment and KLF4 genomic occupancy (GSE57702) over the indicated epidermal genes in differentiated keratinocytes. **G** ChIP-qPCR showing the H3K4me3 epigenetic mark at the indicated epidermal genes loci in differentiated keratinocytes upon NEAT1 depletion (LNA-NEAT1). Average values from *n* = 2 biological replicates measured using three technical replicates are plotted. **H** Detection of the interactions between the SWI/SNF epigenetic factor BRG1 or the epidermal transcription factor KLF4 and NEAT1 by RIP assay. After formaldehyde cross-linking, the coimmunoprecipitated RNAs were quantified by qRT-PCR. Average values from *n* = 2 biological replicates measured using three technical replicates are plotted. **I** ChIP-qPCR showing SFPQ occupancy at the NEAT1 binding site of the indicated epidermal genes loci. Average values from *n* = 2 biological replicates measured using three technical replicates are plotted. Source data are provided as a Source Data file.

decreases H3K4me3 marks on the NEAT1 bound genes KLK5, DSC2 and DLX5 (Fig. 4G).

Since KLF4 has emerged as the top upstream transcriptional regulator of the NEAT1 RNA gene set in differentiated keratinocytes (see Fig. 3) and previous report identified this epidermal transcription factor as a paraspeckles component by a mass spectrometry-based approach[35], we tested whether KLF4 is able to interact with NEAT1 in differentiated keratinocytes. To this aim, we performed RNA immunoprecipitation (RIP) assay and we found that in differentiated keratinocytes KLF4 associates with NEAT1 (Fig. 4H). Intriguingly, in several epidermal genes NEAT1 binding sites overlaps or are in proximity with the KLF4 genomic occupancy (Figs. 4F, S7D, S7E). As control of the RIP assay, we utilized the SWI/SNF catalytic subunit BRG1 (also known as SMARD4), which has been previously demonstrated to be a NEAT1-binding protein and an essential component of paraspeckles assembly and maintenance[36]. Remarkably, BRG1 is also critically involved in the regulation of epidermal differentiation genes and cooperates with KLF4 to induce the expression of epidermal differentiation genes[37–39]. The NEAT1 binding with the paraspeckles component BRG1 together with the evidence that NEAT1 specifically localizes in the paraspeckles in differentiated keratinocytes (see Figs. S3A, 2D), prompted us to test whether paraspeckles proteins are localized over the NEAT1 binding epidermal genes. As shown in Fig. 4I, we found that SFPQ, a paraspeckles component that associates with NEAT1[40], localizes to the promoters of the NEAT1 bound epidermal genes DLX5, KLK5 and DSC2. Furthermore, the DNA binding motif of the transcription factor ZKSCNA3, which has been reported to be associated with the essential paraspeckles component ZNF24[35], is significantly enriched in the top 25% NEAT1 trans genomic sites (Fig. S8). Collectively, these results show that NEAT1, likely in association with paraspeckles proteins, localizes to the TSSs of skin differentiation genes, favoring their efficient expression during the onset of epidermal differentiation.

**NEAT1 depletion affects epidermal differentiation**
To evaluate the physiological relevance of the NEAT1-mediated control of epithelial genes expression, we analyzed the cellular effect of NEAT1 depletion on the differentiation capabilities of primary human keratinocytes. As shown in Fig. 5A, in contrast to MALAT1 depletion, NEAT1 depletion in differentiated keratinocytes impairs the expression of the differentiation markers ZNF750, keratin 10 (K10) and keratin 1 (K1) (Fig. 5A). We confirmed ZNF750 and K10 downmodulation upon NEAT1 silencing by also utilizing a different NEAT1 targeting LNA-Gapmer (Fig. 5B). p63 mRNA levels are not affected by NEAT1 depletion (Fig. 5B). To confirm these data in a more physiological model of skin development, we established an organotypic human epidermal model that recapitulates the gene expression profile and the structure of the human epidermis. NEAT1-depleted epidermis appeared less thick than

the wild-type epidermis, with a substantial reduction of the stratum corneum thickness (Figs. 5C, D). We did not any significant changes of the cell proliferation rate and apoptotic index neither in NEAT1-depleted epidermis nor in NEAT1 depleted keratinocytes in 2D culture (Figs. 5E and S9). NEAT1 silencing by an additional NEAT1 targeting LNA-Gapmer exerted a similar reduction of the stratum corneum thickness (Fig. S10). These phenotypic changes are paralleled by a marked decrease of the expression of both late (loricrin) and early differentiation markers (K10) (Fig. 5F). Conversely, neither the expression of the basal cell markers laminin, p63 and integrin α6β4 nor cell proliferation and cell survival are affected by NEAT1 depletion (Fig. 5F and Fig. S11).

Skin diseases are characterized by alteration of the differentiation program and/or aberrant proliferation of keratinocytes. GO analysis of NEAT1 transcriptome in differentiated keratinocytes indicated that several NEAT1 regulated genes are involved in skin diseases characterized by alteration of keratinocytes differentiation program (see Fig. 3F). Based on this observation, we decided to analyze NEAT1 expression in psoriasis and ichthyosis, two skin diseases characterized by keratinocytes hyper-proliferation and aberrant keratinization, respectively. By analyzing probes matching specific NEAT1 long isoform in GEO datasets, we observed a significant increase of NEAT1_2 expression in samples derived from patients affected by Lamellar Ichthyosis (Fig. 5G). We also observed a similar increase of NEAT1_2 expression in skin samples of ALOX12B knock-out mice (Fig. S12A), a mouse model resembling congenital Ichthyosis[41]. In contrast to Ichthyosis, psoriatic samples are characterized by decrease expression of NEAT1_2, likely reflecting the hyperproliferation status of keratinocytes in this skin disease (Figs. 5G and S12B).

Collectively, these data indicate that NEAT1 is physiologically relevant for the correct activation of the keratinocyte differentiation program and alterations of its expression are found in skin diseases characterized by alteration of proliferation/differentiation balance.

## Discussion
The transcription factor ΔNp63 is a master regulator of epithelial biology and its transcriptional activity is pivotal for regulating epithelial stem cell function and maintaining the integrity of stratified epithelial tissues. So far, the functional link between ΔNp63 activity and long non-coding RNA (lncRNAs) expression and function has not been explored. In this manuscript we report evidence that the lncRNAs NEAT1 is a terminal differentiation-induced lncRNA required for skin differentiation. We found that NEAT1 expression is finely modulated during keratinocytes differentiation and the master epithelial transcription factor ΔNp63 plays a major role on this modulation (see the schematic model in Fig. 5H). In detail, we found that in basal proliferating keratinocytes ΔNp63 is able to repress NEAT1 expression by favoring the HDAC1/2-mediated histone deacetylation on NEAT1 promoter. Upon the onset of keratinocytes differentiation ΔNp63 downmodulation facilitates NEAT1 accumulation and the concomitant assembly of NEAT1 positive paraspeckles in differentiated

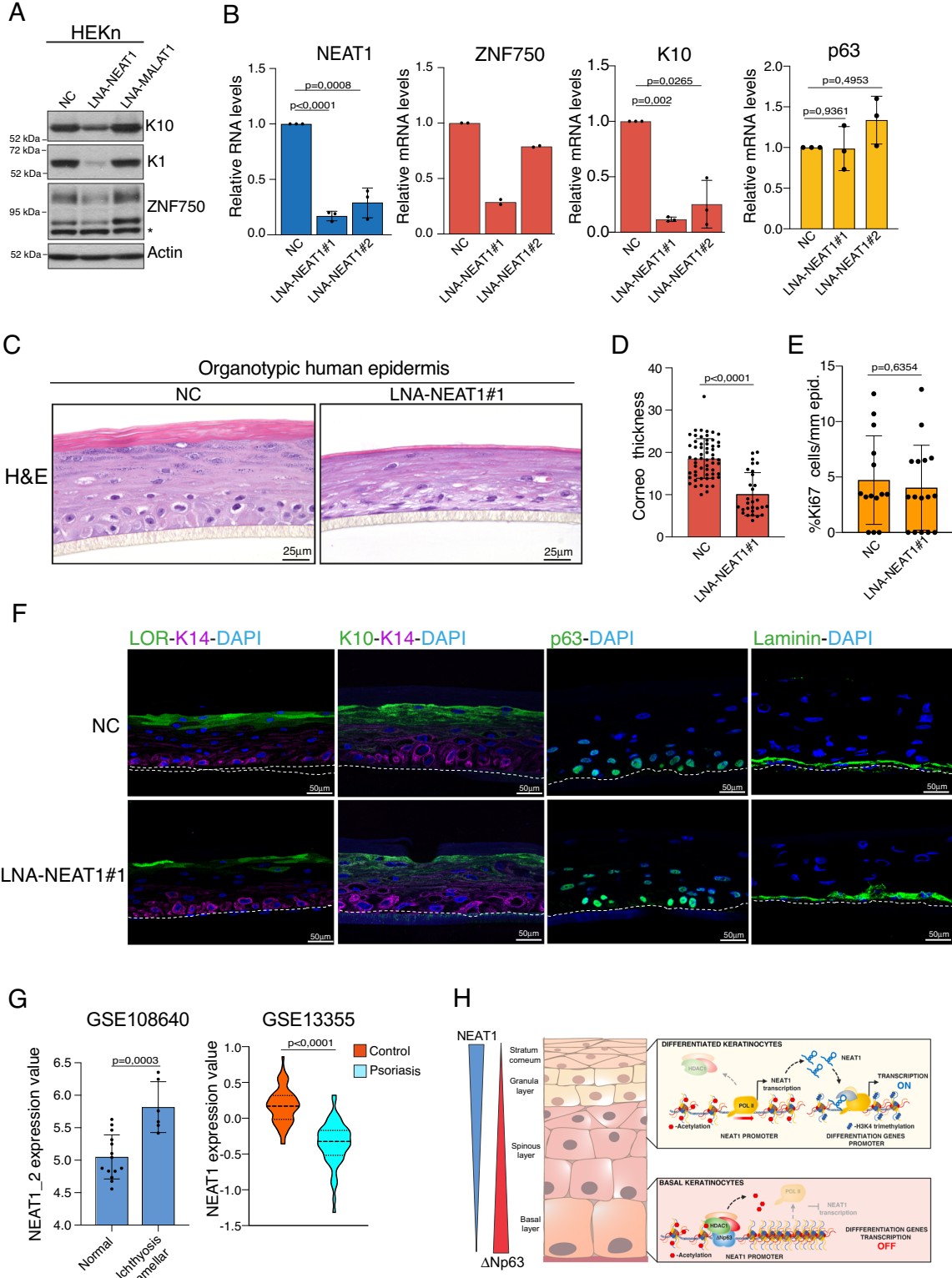

keratinocytes. It is interesting to note that NEAT1 has been previously identified as target of p53[42–44]. Conversely to ΔNp63, p53 induces the transcription of NEAT1 and this circuit is important to mediate the tumor suppressive function of p53[42–44] Therefore, both p53 and ΔNp63 are able to control NEAT1 expression, although in a opposite way.

At functional level, we provide evidence that the differentiation dependent upregulation of NEAT1 might be important to sustain the expression of epidermal differentiation genes. Indeed, RNA-seq analysis associated with global DNA binding profile (ChIRP-seq) revealed that NEAT1 associates with the TSS of key epithelial transcription factors sustaining their expression during epidermal differentiation. NEAT1 binding genes includes epidermal transcription factors (ZNF750, KLF4, ERG3 and DLX5), serine proteases important for corneum integrity (KLK5, KLK6) and components of desmosome (DSC2).

**Fig. 5 | NEAT1 depletion impairs epidermal differentiation. A** Protein levels of epidermal differentiation markers, keratin 10 (K10), keratin 1 (K1) and ZNF750, were quantified by immunoblotting in differentiated keratinocytes transfected with negative LNA GapmeR control (NC), LNA GapmeR NEAT1 (LNA-NEAT1) or LNA GapmeR MALAT1 (LNA-MALAT1). **B** RNA levels of K10, p63 and NEAT1 were quantified by RT-qPCR in differentiated keratinocytes transfected with negative LNA GapmeR control (NC), LNA GapmeR NEAT1#1 (LNA-NEAT1#1) or LNA GapmeR NEAT1#2 (LNA-NEAT1#2). Data shown are the mean of three ($n = 3$) independent biological replicates ± SD. $p$ value was calculated using two-tailed unpaired Student's $t$ test. For ZNF750 mRNA levels quantification, average values from $n = 2$ biological replicates measured using three technical replicates are plotted. **C** Representative image of H&E staining of NEAT1-depleted (LNA-NEAT1) organotypic human epidermis (SKIN 3D) compared to control (NC) organotypic epidermis. The experiment was repeated twice with similar results ($n = 2$). **D** Quantification of stratum corneum thickness in control (NC) and NEAT1-depleted organotypic human epidermis. Data shown are the mean of $n = 59$ (NC) and $n = 29$ (LNA-NEAT1#1) measurements ± SD. $p$ value was calculated using two-tailed unpaired Student's $t$ test. **E** Quantification of Ki67 positive basal cells in control (NC)

and NEAT1-depleted organotypic human epidermis. Data shown are the mean of $n = 14$ and $n = 16$ measurements ± SD for control and NEAT1 depleted epidermis, respectively. $p$ value was calculated using two-tailed unpaired Student's $t$ test. **F** Representative images of the immunofluorescence analysis of control (NC) and NEAT1-depleted (LNA-NEAT1) organotypic skin cultures. Loricrin (LOR) and K10 are markers of differentiated layers, while keratin 14 (K14) and Laminin are markers of the basal layer. Immunofluorescence staining of p63 was used to confirm basal layer. DAPI (blue) was used to visualize nuclei. Dotted lines underline the keratinocyte-fibroblast border. Scale bars, 50 μm. The experiment was repeated twice with similar results ($n = 2$). **G** Analysis of NEAT1 expression in lamellar Ichthyosis (GSE108640) and psoriasis (GSE13355) datasets. In the lamellar Ichthyosis dataset, NEAT1_2 expression value (DNA probe 227062_at) is shown as the mean ± SD of 14 ($n = 14$, normal skin) and 6 ($n = 6$ lamellar Ichthyosis lesions) samples. In the psoriasis dataset, NEAT1 expression value (DNA probe 224565_at) is shown as the mean ± SD of 64 ($n = 64$, normal skin) and 58 ($n = 58$, psoriatic lesions) samples. $p$ value was calculated using two-tailed unpaired Student's $t$ test. **H** Schematic model of ΔNp63-NEAT1 axis function during keratinocytes differentiation. Source data are provided as a Source Data file.

These molecular alterations translate into the inability of NEAT1-depleted keratinocytes to undergo the correct epidermal differentiation program in vitro and in organotypic model of epidermis.

Although we also identified the lncRNA MALAT1 as ΔNp63 transcriptional target whose expression is modulated during skin differentiation similarly to NEAT1, we did not observe any transcriptional dysregulation of epidermal differentiation markers in MALAT1 depleted keratinocytes, suggesting that MALAT1 is not required for the activation of epidermal differentiation program. Although our loss of function approach might not be conclusive on the impact of MALAT1 depletion on epidermal differentiation, several observations are indicative that NEAT1 and MALAT1, although spatially and functionally interconnected, might exert distinct functions. Genetic ablation of MALAT1 did not impact neither global gene expression nor nuclear speckles assembly, nor alternative pre-mRNA splicing in mouse tissues[45]. Conversely, NEAT1 knockout mice, although viable, are characterized by morphogenetic alterations of distinct tissues, some of them related to defects in tissue differentiation[42]. NEAT1 null females indeed display defects in mammary gland morphogenesis and altered expression of genes necessary for *corpus luteum* differentiation[46,47]. More importantly, mice specifically deleted of NEAT1 short isoform do not manifest the phenotypes observed in global NEAT1-deficient mice suggesting that the long isoforms NEAT1_2 and, by extension paraspeckles, are responsible for the defects detected in NEAT1 KO mice[42]. These mice studies also demonstrated that during mammary gland development NEAT1 KO alveolar cells show reduced proliferation rate respect to wild-type cells. We did not observe alteration of cell proliferation in NEAT1 depleted keratinocytes. The discrepancies of the phenotypes observed between human primary keratinocytes and murine alveolar cells could be related to difference of NEAT1 depletion efficiency (KO vs silencing) or can be related to the different cellular context. Accordingly, NEAT1 genetic deletion reduces proliferation of alveolar cells at midgestation (8.5 and 12.5 d post-coitum), but not at the start of pregnancy[47]. Furthermore, difference in mouse and human NEAT1 function and stability may also explain this discrepancy. Indeed, mouse NEAT1_1 and NEAT1_2 isoforms are highly unstable lncRNA, while human NEAT1 is relatively stable[48]. Since many paraspeckles proteins (e.g NONO, SFPQ) contribute to NEAT1 RNA stabilization[49], it is possible that the dynamic of paraspeckles formation and maintenance is different in human and mouse cells. Consequently, NEAT1 depletion could exert different outcomes in human and mouse cells.

It is noteworthy that the link between NEAT1 and cellular differentiation has been described in additional cellular contexts[50]. For instance, NEAT1 expression is upregulated during differentiation of neurons, glia, myeloid cells and muscle, although the molecular details

of its differentiation-mediated regulation have not been elucidated[51–53]. Notably, the functional link between NEAT1 function and cellular differentiation has been also postulated in pathological context. In pancreatic cancer NEAT1 acts as tumor suppressor by regulating the expression of pancreatic differentiation genes[43,54]. More recently, in neuroblastoma the upregulation of NEAT1_2 isoform by morpholino oligo targeting the polyadenylation sites is associated with increased expression of differentiation genes. Based on these observations and our data, we can speculate that NEAT1 and likely paraspeckles might be important determinant of cellular differentiation in different pathophysiological context.

At a molecular level, NEAT1 exploits multiple mechanisms that ultimately impinge upon gene expression. For instance, one of the first function associated with NEAT1-associated paraspeckles, was the ability to modulate the nuclear retention of edited RNAs in paraspeckles[55–57]. NEAT1 is also able to interact with the splicing machinery and microRNA biogenesis apparatus, regulating the maturation of pre-mRNAs and pri-miRNA, respectively[58], or to sequester transcription factors (e.g. SFPQ) impeding their ability to modulate the expression of their target genes[55,59]. By performing global DNA binding profile (ChIRP-seq) in differentiated keratinocytes, we reported that NEAT1 associates with the promoter of key epidermal differentiation genes sustaining their expression during epidermal differentiation. Notably, NEAT1 binding regions of epidermal genes promoter are highly correlated with epigenetic histone modifications, such as histone H3K4 trimethylation, a marker of active transcription chromatin regions. The ability of NEAT1 to bind transcriptional active genes has been also described in other cellular contexts. For instance, in response to estrogen stimulation, NEAT1 re-localizes on the promoter region of estrogen responsive genes and NEAT1 binding region on these promoters is associated with histone marker of active transcription[33]. Intriguingly, similarly to estrogen response, keratinocytes differentiation is characterized by an extensive and substantial transcriptional induction of genes which are crucial for transition from progenitor cells to fully differentiated keratinocytes and for the integrity and maintenance of the epidermis. We can argue therefore that NEAT1 binding to the promoter region can favor the induction of genes in those conditions requiring an extensive transcriptional activation of distinct genes. Since NEAT1 impact on transcription is limited to a specific subset of genes, it is possible that NEAT1 binding to the promoter regions does not imply per se transcriptional activation and it is likely that the NEAT1 effect on specific subset of gene promoters may be dictated by its interaction with specific factors. In line with this assumption, we found that in differentiated keratinocytes NEAT1 binds to the epidermal transcription factor KLF4 which has emerged as the top upstream transcriptional regulator of NEAT1 RNA profile. In

addition to KLF4, we also found that NEAT1 interacts with BRG1, which is an essential component of paraspeckles[36] and cooperates with KLF4 to induce the expression of epidermal differentiation genes[10,38,60]. The association between NEAT1 and paraspeckle proteins in differentiated keratinocytes may also suggest that NEAT1 binding to DNA and its function on epidermal differentiation may be linked to its ability to act as an architectural RNA for paraspeckle assembly. Accordingly, we found that the paraspeckle protein SPFQ is localized to the NEAT1 bound epidermal gene promoters. However, we can rule out the possibility of a direct association of NEAT1 with DNA since a previous report demonstrated the ability of NEAT1 to bind the DNA in the absence of proteins[61]. Along NEAT1 sequence there are indeed significant triplex-forming regions (TFR) with the ability to interact with specific DNA sequence[61]. However, this study has been performed in vitro and it is not clear whether a similar scenario occurs in vivo.

Collectively, these data suggest a model in which NEAT1, likely in association with paraspeckles proteins, facilitates the recruitment of critical epidermal transcription factors on differentiation gene promoters promoting thus the activation of the differentiation program.

Based on the impact of NEAT1 dysregulation on skin differentiation, we also investigated whether NEAT1 levels are altered in those skin diseases characterized by alteration of differentiation/proliferation pathway. We found that NEAT1 is upregulated in Ichthyosis lamellar human samples and in Alox12B KO mice, a mouse model resembling this skin disease. Ichthyosis is skin diseases characterized by increased differentiation and abnormal keratinization. Conversely, in psoriasis, a skin hyperproliferative disorders, we observed a marked reduction of NEAT1 levels.

In conclusion, our results unveiled the lncRNA NEAT1 as an additional player of the highly complex regulatory network by which the master epidermal transcription factor ΔNp63 controls epidermal homeostasis, which might be crucial for our comprehension of the mechanisms underlying epidermis development and skin diseases.

## Methods

### Cell lines and treatment

A253 (submaxillary salivary gland carcinoma, male, ATCC HTB-41) cells were cultured in McCoy's medium (Gibco, Invitrogen); FaDu (Pharynx squamous cell carcinoma, male, ATCC HTB-43) were grown in Eagle's minimum essential medium (EMEM) (Lonza); HCC1954 (human breast ductal carcinoma, ATCC CRL-2338) cells were grown in RPMI-1640 medium (Gibco, Invitrogen); All media were supplemented with 10% fetal bovine serum (FBS), 100 μg/ml penicillin and 100 μg/ml streptomycin (Gibco, Invitrogen). hMEC (Human primary mammary epithelial cells, ATCC PCS-600-010) were grown in Mammary epithelial cell complete medium (Mammary epithelia Growth kit PCS-600-040, ATCC), HEKn (Normal Human epidermal keratinocyte, neonatal, Thermo-Life Technologies, cat.: C0015C) were cultured in EpiLife medium with addition of Human Keratinocyte Growth Supplements (HKGS, Life Technologies). Cells were routinely tested for mycoplasma contamination by MycoAlert mycoplasma detection kit (LONZA LT07-418). Cells were cultured at 37 °C with 5% CO2. Differentiation was induced by adding 1.2 mM CaCl2 to the culture medium of sub confluent HEKn. The inhibitor of Histone deacetylase (ITF2357, Givinostat) was added to cell medium to a final concentration of 1, 5 and 10 μM for 12 h.

### Transfection and BrDU analysis

siRNA oligos and Antisense LNA GapmeRs transfection, were transfected using Lipofectamine RNAiMAX (Thermo Fisher) according to the manufacturer's instructions. siRNA oligo specific for p63 (sip63) and ΔNp63 (siΔNp63) mRNAs and nonrelevant siRNA (SCR) were purchased by Sigma-Aldrich. LNA GapmeRs oligos against NEAT1 ncRNA (LNA-NEAT1#1 and LNA-NEAT1#2), MALAT1 ncRNA (LNA-MALAT1) and non-relevant gene (NC) were generated by Qiagen

(Exiqon). The siRNA oligos and Antisense LNA GapmeRs sequences are shown in Table S2.

Incorporation of BrdU during DNA synthesis was evaluated with the Click-iT EdU Flow cytometry assay kit, following the manufacturer's protocol (Invitrogen). Cell cycle was analyzed using cytoFLEX flow cytometer (Beckman Coulter Life Sciences). Twenty thousand events were evaluated.

### RNA Isolation and RT-qPCR

Total mRNA was isolated using the RNeasy mini kit (Qiagen) following the manufacturer's recommendations, quantified using a NanoDrop Spectophotometer (Thermo Scientific) and retrotranscribed by SensiFast cDNA Synthesis (Bioline), according to the manufacturer's protocol. Real-time PCR was performed using SYBR-Green PCR Master Mix (Promega), with the QuantStudio 5 Real-Time PCR Systems (Applied Biosystems). The expression of each gene was defined from threshold cycle (Ct), and the relative expression levels were calculated using the 2-ΔΔCt method. Primer efficiencies (PE) were evaluated by performing qPCR of the target assay on 10-fold dilution series. The slope of the standard curve was translated into an efficiency (e) value with the following equation:

$$PE(\%) = \left( -\frac{1}{10e^{slope}} - 1 \right)$$

The primers utilized for RT-qPCR are shown in Table S3.

### Immunoblotting

Immunoblot analysis was performed using A253, FaDu, HCC1954 and hMEC cell extracts obtained by lysing cell pellets with Triton buffer (50 mM Tris-HCl pH 7.5, 250 mM NaCl, 50 mM NaF, 1 mM EDTA pH 8, 0.1% Triton), supplemented with protease inhibitors (Roche), DTT, PMSF, and sodium orthovanadate (NEB). HEKn cells were lysed in SDS lysis buffer (100 mM Tris, pH8,8, 1%SDS, 5 mM EDTA, 20 mM DTT and 2 mM AEBSF). Proteins were separated by SDS/PAGE, transferred onto PVDF membranes, and blocked with PBS-T (phosphate-buffered saline and 0.1% Tween-20) containing 5% nonfat dry milk for 1 h at room temperature (RT). The incubation with primary antibodies was performed for 2 hrs at RT, followed by incubation with the appropriate horseradish peroxidase-conjugated secondary antibody (Biorad anti-mouse cat. num. 170-5047, dilution: 1:6000; Biorad anti-rabbit cat. num. 170-6515, dilution: 1:6000). Detection was performed with ECL Western Blot Reagent (Perkin-Elmer). We utilized the following antibodies: rabbit monoclonal anti p63-α (Cell Signaling Technology, clone D2K8X, dilution: 1:1000), rabbit polyclonal anti-KRT1 (BioLegend, PRB-149P, dilution: 1:1000), rabbit polyclonal anti-KRT10 (Covance PRB159P, dilution: 1:5000), rabbit polyclonal anti-ZNF750 (Sigma-Aldrich HPA023012, dilution: 1:200), rabbit polyclonal anti-PARP (Cell Signaling #9542, dilution: 1:1000) and mouse monoclonal anti β-actin (Sigma-Aldrich, #AC-15, dilution: 1:50000).

### Chromatin Immunoprecipitation (ChIP) Analysis

ChIP experiments were performed using the MAGnify Chromatin Immunoprecipitation System (Invitrogen, 492024) by following manufacturer's protocol. Chromatin was sonicated at 25% amplitude for 30 min (20″ sonication/30″ pause) by Bioruptor UCD-200 (Diagenode). The chromatin extract was incubated with Dynabeads Protein A/G coupled to specific antibody or negative control overnight at 4 °C. The immune complexes were washed and treated with proteinase K (20 mg/mL) at 55 °C for 15 min to reverse the cross-linking. DNA was isolated with DNA purification magnetic beads and used for PCR analysis. PCR reactions were performed by using GoTaq G2 Flexi DNA polymerase (Promega) according to manufacturer's protocol. The PCR products were analyzed by electrophoresis on agarose gel. qRT-PCRs were performed by using SYBR-Green PCR Master Mix (Promega). The

Primers used are listed in Table S2. The antibodies used were as follows: rabbit anti-p63α (D2K8X; Cell Signaling), rabbit anti-HDAC1 (ab19845 Abcam), rabbit Anti-Histone H3 (acetyl K9 + K14 + K18 + K23 + K27) (ab47915 Abcam), rabbit anti-SFPQ (Sigma-Aldrich PLA0181), rabbit anti-histone H3 (tri methyl K4) (ab8580 Abcam), rabbit IgG and mouse IgG (Invitrogen, 492024) (negative control).

## Cross-linked RNA Immunoprecipitation (RIP)
RIP assay was performed following the protocol described by[62] with minor modifications. Briefly $10^7$ differentiated HEKn were cross-linked adding to cell suspension the necessary volume of formaldehyde to have 1% final concentration and incubating 10 min at room temperature. Cells were resuspended in cell lysis buffer (10 mM Tris-HCl pH 7.4, 10 mM NaCl, 0.5 % NP-40 supplemented with PMSF, protease inhibitors (Roche), 1 mM DTT and Superase-in (20U/ml)) and nuclei were isolated by passing cells suspension through a 29-gauge needle 10 times. Nuclei were resuspended in nuclei resuspension buffer (50 mM HEPES-NaOH pH 7, 10 mM MgCl 2 supplemented with PMSF, protease inhibitors (Roche), 1 mM DTT and Superase-in (20U/ml)), sonicated for 30 min (30″ sonication/30″ pause) by Bioruptor UCD-200 (Diagenode) and then incubated with 15 U of TURBO DNase (Thermo Fisher). Cell lysates were incubated overnight with anti-KLF4 (AF3640, RD Systems), anti-BRG1 (A300-813A Bethyl Laboratories) or rabbit IgG (Invitrogen) and immunocomplexes were isolated with protein-A Dynabeads (Thermo Fisher). RNA was purified with TRIzol (Thermo Fisher), retrotranscribed with SuperScript VILO Master Mix (Thermo Scientific) and then analyzed by qPCR.

## Organotypic epidermis culture
Organotypic human epidermis were generated using 3D full thickness starter kit from CELLnTECH following the manufacturer's protocol. Briefly HEKn were transfected with NC, LNA-Neat1#1 or LNA-Neat1#2 Antisense LNA GapmeR and 24 h after transfection were seeded on top of the established dermal fibroblast layers, previously arranged into PET inserts starting from normal human fibroblast (Cascade Biologics,Thermo Fisher). After three days, the inserts were transferred into new plates containing 2.5 mm thick spacers and the models were grown to the air-liquid interface for 12 days before being fixed with 4% paraformaldehyde overnight at 4 °C, processed and paraffin-embedded.

## RNA FISH
RNA FISH experiments were performed using Stellaris® RNA FISH kit (Biosearch Technologies) and following the manufacturer's recommendations. Proliferating and differentiated HEKn were cultured on coverslips and were fixed in 4% paraformaldehyde for 10 min and then permeabilized with 0.1% Triton X-100 for 10 min. Incubation with the human NEAT1 probe with Quasar 570 Dye or human MALAT1 probe with Quasar 570 (Biosearch Technologies, Inc.) for 4–16 h at 37 °C in a dark humid chamber. Was added to 1 µg/mL 4′,6-diamidino-2-phenylindole (DAPI; Sigma) for nuclear DNA staining. Coverslips are placed on a microscope slide with ProLong Gold Antifade (Thermo Fisher Scientific) and images acquired on a confocal microscope Leica Stellaris 8.

## Immunofluorescence (IF) and TUNEL assay
Paraffin-embedded slides of organotypic human epidermis were incubated at 58 °C for 1 h, dewaxed by Bio-Clear washing (Bio-Optica) and rehydrated by serial dilution of ethanol (100, 90, 80, 70 and 50% ethanol). Antigen retrieval was performed by boiling in 0.01 M Sodium Citrate Buffer pH 6.0. IF staining was performed as follows: 1 h blocking in 5% goat serum (Gibco) in PBS at RT; 2 hrs primary antibody incubation at RT; and 1 h secondary antibody and DAPI (4′,6-diamidino-2-phenylindole) incubation at RT. The following antibodies were used: mouse polyclonal anti-KRT10 (Covance PRB159P, dilution: 1:1000); mouse monoclonal anti p63-α (AB735 Abcam, dilution: 1:100), rabbit polyclonal anti-Loricrin (Covance PRB145P, dilution: 1:1000); rabbit

polyclonal anti-KRT1 (BioLegend, PRB-149 P, dilution: 1:1000); rabbit anti-laminin (Sigma, L9393, dilution: 1:200), mouse anti-integrin α6β4 (BD, 611232, dilution: 1:200), rabbit anti-Ki-67 (Cell Signaling D3B5, dilution: 1:200), goat anti-rabbit 488 (LifeTechnologies, A11034, dilution: 1:1000); goat anti-mouse 488 (LifeTechnologies, A28175, dilution: 1:1000); goat anti-mouse 568 (Life Technologies, A11019, dilution: 1:1000).

TUNEL assay was performed with the Click-iT Plus Tunel Assay, following the manufacturer's protocol (Invitrogen). Positive controls were prepared incubating cells with DNase (RNase-Free DNase Set, Qiagen) for 30 min at room temperature. Images were acquired by confocal microscope Leica Stellaris 8 or Nikon A1 (Nikon NIS elements software)

## Sequential IF and RNA FISH
Sequential IF and FISH experiments were performed for cells and tissue sections. Paraffin-embedded slides were incubated at 58 °C for 1 h and then followed the IF protocol, as described above. While, HEKn were grown on slides and fixed in 4% paraformaldehyde, permeabilized with 0.1% Triton X-100 for 10 min. IF staining was performed as follows: 1 h blocking in 5% goat serum (Gibco) in PBS at RT, 2 hrs primary antibody incubation at RT; 1 h secondary antibody and DAPI (4′,6-diamidino-2-phenylindole) incubation. We procced with another fixation step to start the FISH procedure for both tissue sections and cell slides. Briefly, slides were incubated with the human NEAT1 probe with Quasar 570 Dye or human MALAT1 probe with Quasar 570 (Biosearch Technologies, Inc.) for 4–16h at 37 °C in a dark humid chamber. Was added to 1 µg/mL 4′,6-diamidino-2-phenylindole (DAPI; Sigma) for nuclear DNA staining. Coverslips are placed on a microscope slide with ProLong Gold Antifade (Thermo Fisher Scientific) and images acquired on a confocal microscope Leica Stellaris 8.

The following antibodies were used: rabbit anti-SFPQ (Sigma-Aldrich PLA0181, dilution: 1:200), mouse monoclonal anti p63-α (AB735 Abcam, dilution: 1:100). The following probes were used: human NEAT1 with Quasar 570 Dye and human MALAT1 with Quasar 570 Dye (Biosearch Technologies).

## Haematoxylin/eosin staining
Paraffin-embedded sections of organotypic human epidermis were dewaxed and rehydrated as described before[5], then incubated 5 min in Mayer's haematoxylin (Bio-Optica) solution, extensively washed in distilled water, incubated 5 min in Eosin Y alcoholic solution (Bio-Optica), extensively washed in distilled water and finally dehydrated by 70, 90, 100% ethanol solution incubation. Slides were mounted using Bio Mount HM (Bio-Optica). Images were acquired on a microscope Leica DM6.

## ChIRP-seq
Chromatin isolation by RNA purification (ChIRP) assay was performed following the protocol described Chu et al., with minor modifications. Briefly, $2 \times 10^7$ differentiated primary keratinocytes (4 days in $CaCl_2$ containing medium) were cross-linked for 10 min at room temperature with 1% glutarldheyde (Sigma-Aldrich). Cells were lysed in Lysis Buffer (50 mM Tris-HCl, pH 7.0, 10 mM EDTA, 1% SDS) supplemented with PMSF, protease inhibitors (Roche), 1 mM DTT and Superase-in (20U/ml). The chromatin was then sheared using sonicator (Bioruptor UCD-200; Diagenode) at 4 °C for 3 h. The chromatin extract was incubated with biotinylated NEAT1 DNA probes at 37 °C for 4 hrs with shaking. We utilized the following mix of NEAT1 antisense probes: NEAT1 CO1: TTCCTTCTCGCACCCC-CAGC/iSp18//3BioTEG; NEAT1 CO2:TGTCTGTCCCCTGAAGCCCTG/ iSp18//3BioTEG; NEAT1 CO3: CTAGCCACTTCCTCCCCCACAA/iSp18// 3BioTEG. As control we utilized the following mix of LACZ DNA probes: LacZ_1 TCACGACGTTGTAAAACGAC/iSp18//3BioTEG), LacZ_2 GCTGATTTGTGTAGTCGGTT/iSp18//3BioTEG and LacZ_3

TTTACCTTGTGGAGCGACAT/iSp18//3BioTEG. The chromatin complexes were purified using Dynabeads MyOne Streptavidin C1 (Thermo Fisher Scientific). DNA was isolated with phenol-chloroform and used for qPCR analysis and preparation of high-throughput sequencing libraries (Accel-NGS 2 S Plus DNA Library Kit, Swift Biosciences) per Illumina protocol.

ChIRP-seq data were analyzed with "chipseq" version 1.2.2 pipeline included into the nf-core platform (https://nf-co.re/chipseq) with default parameters (doi: 10.5281/zenodo.3240506). Visualization of peak profiles were obtained using bigwig files of each experiment provided as input to Integrative Genomics Viewer (IGV)[63]. ChIP-seq of H3K4me3 (GSE98483) on primary keratinocytes in day 4 of differentiation data were downloaded from the Gene Expression Omnibus. ChIP-seq of H3K27me3 (GSE175176) on primary keratinocytes in day 3 of differentiation data were downloaded from the Gene Expression Omnibus (ENCODE project). Intersection and visualization of genomic region was performed with Intervene software[64]. Functional interpretation of cis-regulatory regions was performed with Regions Enrichment of Annotations Tool (GREAT)[65].

To perform gene group analysis, the NEAT1 binding site was partitioned based on the relative enrichment over background and the top 25%, top 50%, and all sites were selected, respectively. Subsequently, the frequency of NEAT1 binding at each gene under consideration was calculated.

The putative binding motifs in NEAT1 were determined using the MEME algorithm[66], with a focus on the top 50% and top 25% NEAT1.

### RNA-seq and bioinformatic analyses
Total RNA was extracted using the RNeasy mini kit (Qiagen) and purified from DNA contamination through a DNase I (Qiagen, IT) digestion step. Quantity and integrity of the extracted RNA were assessed by NanoDrop Spectrophotometer (NanoDrop Technologies, DE) and by Agilent 2100 Bioanalyzer (Agilent Technologies, CA), respectively. RNA libraries for sequencing were generated in triplicate using the same amount of RNA for each sample according to the Illumina TruSeq Stranded Total RNA kit with an initial ribosomal depletion step using Ribo Zero Gold (Illumina, CA). The libraries were quantified by qPCR and sequenced in paired-end mode (2 × 100 bp) with NovaSeq 6000 (Illumina, CA). For each sample generated by the Illumina platform, a pre-process step for quality control was performed to assess sequence data quality and to discard low-quality reads.

RNA-seq data were analyzed with "rnaseq" version 3.3 pipeline included into the nf-core platform (https://nfco.re/rnaseq) with default parameters (doi:10.5281/zenodo.5550247). Differential expression analysis was carried out using DESeq2 package[67]. Normalized counts were expressed as the variance stabilizing transformation (VST function). Enrichment analysis, Disease Ontology geneset interpretation were performed with the ShinyGO version 0.76[68]. Unsupervised hierarchical clustering was performed with the ComplexHeatmap R package[69]. Volcano plot was generated with EnhancedVolcano R package. Expression data of primary human keratinocytes in high calcium differentiation conditions were downloaded from the Gene Expression Omnibus (GSE18590). Enrichment analysis with the intersection of GEO dataset was performed with Gene Set Enrichment Analysis (GSEA) software. Trancription factor analysis was performed by DoRothEA software[70].

The level of significance was defined as $p < 0.05$. Statistical analyses were carried out using R software. Transcriptional regulators of siNEAT1 RNA-seq were identified using IPA tool and sorted by the activation z-score[71]. RNA-seq data have been deposited in the NCBI's Gene expression Omnibus (GSE205960).

### LncRNAs array and data analyses
hMEC, HCC1954, A253, FaDu and NHEK cells were transfected with scramble (SCR) or p63 siRNA (sip63). 48 h after transfection, total RNA

was extracted and quantified as described. LncRNA Expression profiling analysis of the cells was performed using an in-house constructed microarray (University of Texas MD Anderson Cancer Center, platform ID: GPL22858). Array was scanned on Agilent SureScan Microarray Scanner (Agilent Technologies G2600D SG12434244). Data pre-processing steps of background-correction, normalization and summarization were performed in R using functions in Limma library. The raw intensity for each probe is the median feature pixel intensity with the median background subtracted. Data was quantile normalized followed by log2 transform. Signals from probes measuring same ncRNA were averaged. Statistical significance was defined as a $p$ value < 0.05 and fold change in absolute value > 1.1. Probes differentially expressed between samples were thus identified. The analysis of the microarray data indicating the name of the lncRNAs probe sequence, mean value of each condition, FCH, $p$ value and FDR for each cell lines have been included in the source data file. LncRNA microarray data have been deposited in the NCBI's Gene expression Omnibus (GSE232654).

### NEAT1 expression in skin diseases
NEAT1_2 expression levels in Ichthyosis lamellar (GSE108640), were extrapolated by downloading the expression value refereed to specific probe (238320 or 227062) mapping NEAT1_2 long isoform. The same method has been applied for evaluating NEAT1_2 expression in psoriasis (GSE13355) and in ALOX12B knock-out mice (GSE127435). Data were plotted as mean ± SD and $p$ value was calculated by Student's $t$ test. Differences were considered statistically significant at $p < 0.05$.

### Statistical information
For RT-qPCR studies, the expression of each gene was defined from threshold cycle (Ct), and the relative expression levels were calculated using the 2-ΔΔCt method. The number of biological replicates per experiment varied and it is mentioned in the figure legends. Generally, the relative expression levels of indicated genes were displayed as bars representing the mean of three biological replicates ($n = 3$) ± SD. The number of biological replicates per experiment varied and it is mentioned in the figure legends. ChIP-assay data were plotted as average of two independent biological replicates. The type of statistical test utilized to calculate the $p$ value was reported in the figure legends. When appropriate, the exact $p$ value was reported in the figures.

### Reporting summary
Further information on research design is available in the Nature Portfolio Reporting Summary linked to this article.

## Data availability
All sequencing data that support the findings of this study have been deposited in the National Center for Biotechnology Information Gene Expression Omnibus (GEO) under the Series accession number GSE205961. This Series includes the access to RNA-seq data (GSE205960) ChIRP-seq data (GSE205959) and lncRNAs microarray (GSE232654). Public ChIP-seq was downloaded from GEO database under the accession number GSM1446927. Public RNA-seq data were downloaded from GEO database under the accession numbers GSM1446880 and GSM1446883. Public lamellas Ichthyosis and psoriasis microarray data were downloaded from GEO database under the accession numbers GSE108640 and GSE13355, respectively. Public ALOX12B knock-out mice microarray data were downloaded from GEO database under the accession number GSE127434. Source data are provided with this paper.

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

## Acknowledgements

The authors thank Prof. Francesca Bernassola for critical suggestions. This work has been supported by the Associazione Italiana per la Ricerca contro il Cancro (AIRC) to A.P. (IG#24678), PRIN2017XCXAFZ to A.P. This work has been also partially supported by the Lazio Innova Progetto Gruppo di Ricerca 2020 A0375-2020-36585 to A.P.

## Author contributions

A.P. conceived the project. A.P. and V.G. wrote the manuscript. G.M. oversaw the study and provided advice. V.G. and C.F. performed most of the experiments. In detail, C.F. performed the experiments described in Fig. 1, Fig. 2, Fig. 5A-B, Fig. 5F-G, Fig. S1A, S1C, S2A-D, S3A, S3C, S4A and S12. V.G. performed experiments described in Figs. 4G-I, S2E-F, S6A, S7C and S10. V.L.B. performed 5D, 5E, S1C, S3B, S4B, S9, S10, S11. S.D.D. and V.G. performed organotypic epidermal model (Fig. 5C-D and S10); A.M. and M.M. performed organotypic skin section. S.S. and G. Co. performed the bioinformatic analysis (Fig. 3, Fig. 4B–F, Figs. S5, S6, S7A, S7B, S7D, S7E, S8). F.D. and M.F. performed the RNA-seq library construction and ChIRP protocol setup, respectively. G.Ca. performed the lncRNAs-based array. All authors have approved this submitted version.

## Competing interests

The authors declare no competing interests.
