## [Peer Review File · Nature Communications]

The long non-coding RNA NEAT1 is a Δ Np63 target gene modulating epidermal differentiationREVIEWER COMMENTS

Reviewer #1 (Remarks to the Author):

In the manuscript "dNp63-mediated repression of the lncRNA NEAT1 modulates epidermal differentiation", the authors use a combination of knock-down, expression profiling, ChIP-qPCR, ChIRP and bioinformatic approaches, as well as FISH, IF and 3D epidermal reconstruction assays to study the role of dNp63 in regulating NEAT1 and MALAT1 expression and the role of NEAT1 in human keratinocyte differentiation and diseased states.

The experiments are generally well executed and presented in clear and mostly compelling figures. However, the connection between the LNA-NEAT1 RNA-seq and NEAT1 ChIRP results as presented is based on a few selected genes and currently provides limited insight into the general role of NEAT1 in epidermal biology. In addition, some of the other conclusions may be strengthened by providing additional controls (experimental and/or computational).

Major comments:

1) The introduction lacks any mention of the known roles of lncRNAs in human epidermal biology. As the first experiment the authors present is a (targeted) expression profiling of lncRNAs after dNp63 knock-down, a section on what is currently known about lncRNA expression and function in human keratinocytes is warranted and should be included to provide context to the work presented in this manuscript.

2) As NEAT1 expression was not covered by the custom lncRNA array, it would be of interest, yet most of the work presented in the manuscript is on NEAT1 regulation and function, it would be of interest to add a figure on NEAT1 RT-qPCR after dNp63 knock-down in the 5 different cell lines (Figure S1). Especially as the justification for focussing on MALAT1 from this analysis is that it is upregulated after dNp63 knock-down in all 5 lines.

3) The authors show predominant (potentially exclusive? Although this is difficult to assess from the provided images) localisation of NEAT1 in paraspeckles in differentiated keratinocytes. Do the authors think that its function and therefore the effect of NEAT1 LNA-mediated knock-down is associated with this localisation?

In figure 4, the authors present ChIRP data to investigate NEAT1 genomic localisation and the potential consequences on transcription regulation of these regions. Does this suggest a model where NEAT1 'draws' genomic regions into paraspeckles to regulate their expression?

At the moment these observations seem unconnected and it is not clear how paraspeckle localisation of NEAT1 is important for keratinocyte biology beyond the observation that differentiated cells contain paraspeckles and proliferating cells do not and how it ties in with the role of NEAT1 on the genome.

4) The authors present ChIRP data on the genomic localisation of NEAT1 in differentiated keratinocytes using a pool of 3 antisense capture oligos (COs). As a control they use the sense version of 1 of these COs. When interrogating the data (bigWig and called peak bed files downloaded from GEO using the provided reviewer token) it seems that ~45% of the 84,651 peaks called from the antisense probe ChIRP signals overlap with peaks (total of 120,429 peaks) called from the antisense (control) probe signals. Moreover, in the genome-browser (viewing the bigwig files) when there is signal in the antisense track, there usually is signal in the sense (control track), albeit weaker as indicated by the authors in the manuscript (supplemental figure S6), which does overlap with H3K4me3 ChIP-seq signal.

Can it be excluded that these differences in signal may have arisen from the fact that the authors used 3 antisense COs and 1 sense CO?

Can the authors exclude a contribution of non-specific binding of the probes to actively transcribed open chromatin regions?

Are the NEAT1 binding regions enriched in specific DNA sequence motifs and how do they relate to the

COs used in the ChIRP assay?

The authors may need to include more substantial controls to convince the eventual readers of the strength of the ChIRP dataset and the conclusions drawn based thereon. For instance, ChIRP after LNA-NEAT1 transfection, ChIRP with an orthogonal set of COs. These may be done using sequencing as a read-out, but potentially a qPCR read-out for representative binding sites may already suffice. Alternatively, the authors can choose to explicitly state the limitations of the dataset as generated and presented currently in the manuscript text and discuss the impact on the strength on the conclusions that can be drawn at this stage.

5) The authors show an overlap between NEAT1 ChIRP signals and a small selected set of NEAT1 dependent differentiation associated genes (Figure 4F). It is unclear from the presented analyses whether this proposed regulation of differentiation genes by NEAT1 binding is a general principle or not. As the authors have a complete set of NEAT1 responsive genes (Figure 3) as well as genome-wide NEAT1 binding data, it would be of interest to include a more global analysis of the claim that NEAT1 controls differentiation genes. Some specific questions relevant to this are: What globally happens to the expression of NEAT1 bound genes during differentiation and upon LNA-NEAT1 silencing? What are the proportions of genes whose expression goes up/down. Are these enriched in genes involved in specific processes (GO term analysis)? What about the overlap of these NEAT1 bound genes with dNp63 (repressed) target genes? Providing more body to the analyses of the ChIRP data will help assess the strength of the conclusions currently drawn by the authors.

6) The 3D organotypic human epidermal equivalent experiments are non-trivial and well executed. The effect on epidermis thickness is convincing. Moreover, it seems that not only the thickness of the cornified layer is affected, but also that the granular layer is absent (based on morphology and lack of the typical granulated nuclei in the H&E stained sections in Figure 5C). There does seem to be some discrepancy between the LNA-NEAT1#1 and LNA-NEAT1#2 in terms of ZNF750 expression. Have the authors confirmed the effect on cornified layer thickness using the second LNA as well?

7) The increased and decreased level of NEAT1 expression in Ichthiosis and Psoriasis, respectively, may reflect changes in ratios of non-differentiated versus differentiated cell populations in these diseases, rather than a specific association of NEAT1 with these afflictions. For instance, do other (classical) markers of differentiation show a same/similar association?

Minor points:

1) As written now, the title grammatically seems to suggest that the regulation of NEAT1 expression by dNp63 takes place during differentiation. However, the authors convincingly show that the regulation takes place, in the form of HDAC-mediated repression, in proliferating (non-differentiated) cells. Therefore, the title does not seem to represent the conclusions in the most intuitive way.

2) In the Methods section, the description of the differentiation induction does not include at which level of confluency of the culture the CaCl₂ was added. This is a key parameter in these assays and should be included.

3) What was the knock-down efficiency of NEAT1 and MALAT1 in the samples used for RNA-seq analysis described in Figure 3. Are these the same samples as depicted in Figure 5A? If so, please state. If not, please indicated knock-down efficiency.

4) In figure 4 the authors show overlap between NEAT1 binding regions and H3K4me3 marked transcription start sites, suggesting that these NEAT1 bound genes are actively transcribed. Did the authors extend these analyses to investigate a quantitative relationship between NEAT1 signal and gene expression level (eg steady state RNA abundance by RNA-seq)?

Reviewer identity: Klaas Mulder

Reviewer #2 (Remarks to the Author):

This study nicely describes the novel observation that the NEAT1 lncRNA controls epidermal differentiation. The manuscript is thoughtful, clear and well written with logical flow and clear description of rationale and experiments. The study begins with a lncRNA library screen to identify RNAs that are regulated by the epidermal transcription factor deltaNp63. However, NEAT1 is not included in the library, so does not factor in the screen. The authors focus on a different RNA MALAT1, that is regulated by p63 but ultimately has no role in controlling differentiation. The link between this work and the authors switch to working on NEAT1 is unclear. Why was it not included in the initial screen, and how was it chosen for follow up without that data?

Regardless, the authors show comprehensively that NEAT1 expression is repressed by deltaNp63 mediated recruitment of HDACs, and that concomitant with differentiation and loss of deltaNp63, NEAT1 levels increase. They show conclusively that NEAT1 plays a role in differentiation and induction of epidermal gene expression, by localization to target gene promoters, and loss of their expression upon NEAT1 knockdown. Finally, the authors propose a role for NEAT1 activity in epidermal disease, as an increase in NEAT1 expression is seen in the hyper differentiation disease Ichthyosis. This link is more tenuous and the authors do not show data describing overexpression of NEAT1 in their model system. If they wish to make this claim they should show data describing the overexpression phenotype. If NEAT1 is overexpressed in keratinocytes can it drive differentiation? Similarly, is there a feedback loop between NEAT1 and deltaNp53? Can NEAT1 expression repress deltaNp53 expression? Finally is it not clear how NEAT1 is impacting gene expression. The authors show its localization to the TSS of epidermal genes, but what is the proposed model for its activity there? Is it acting as a scaffold for RNA binding proteins that promote transcription?

This work is a solid contribution to the field and provides a novel component to the control of epidermal differentiation. As such I approve its publication in nature communication after the issues described above are addressed.

Reviewer #3 (Remarks to the Author):

In this manuscript, Fierro et al., studied the regulation of NEAT1, a well-known long-noncoding RNA, by deltaNp63, and the function of NEAT1 in the differentiation of human keratinocytes. The major findings are the repression of NEAT1 by deltaNp63 through the recruitment of HDAC in proliferative keratinocytes, the function of NEAT1 in promoting epidermal differentiation, and the direct binding of NEAT1 on the promoters of several key regulators of epidermal differentiation, including ZNF750, KLF4 and DLX5. Overall, this study is generally well done and provides interesting insights into the regulation and function of NEAT1 in human epidermal differentiation. I have the following comments that should be addressed by the authors.

Major points:

1. ChIP assays, including both ChIP-seq and ChIP-PCR, could suffer from non-specific crosslink of TF target to DNA sequences, in particular the ones close to the TSS. To firmly establish the direct binding of p63 to the binding sites on MALAT1 and NEAT1 loci, they should identify the canonical p63 motif within the peak. In addition, it is intriguing that p63 negatively regulates MALAT1 and NEAT1 through the recruitment of HDAC to these sites. It will be interesting to identify the determinant for the activation vs repression function of p63 e.g. in which context p63 recruits HDAC and in which context p63 recruits pol II? If they clone the binding site and perform promoter/enhancer assays in the same

cells, will they still observe negative regulation? These additional studies can strengthen the proposed mechanism mediated by p63.

2. Previous studies (Standaert et al., RNA 2014; Adriaens et al., RNA 2019) have shown that genetic KO of NEAT1 (both NEAT1 and NEAT1_2) causes reduced proliferation during mammary gland development and NEAT1 (the short isoform) is dispensable for the function. Although it is possible that mouse and human NEAT1 function may be different, they should carefully address the potential differences. At minimum, they should check whether NEAT1 KD by LNA can alter cell proliferation in raft culture experiments in Fig. 5. And they should discuss how mouse NEAT1 appears to be required for cell proliferation in mammary gland whereas human NEAT1 appears to promote epidermal differentiation.

3. They used ChIRP-seq to identify NEAT1 associated DNA sequences and identified the binding of NEAT1 to the promoters of several important differentiation genes, such as ZNF750, KLF4 and DLX5. They should distinguish whether such binding is mediated by NEAT1 (short isoform) or NEAT1_2 (long isoform). To do so, they should first distinguish the ratio between the short and long isoforms, based on RNA-seq data. If there are significant portions of short and long isoforms, they may need to revise their ChIRP-seq approach since all 3 probes bind to the shared 5' regions of both short and long isoforms. Fundamentally, they need to probe deeper for the mechanism of how the binding of NEAT1 on the promoter of these genes promotes their expression. Their current data only show that NEAT1 bound promoters have enrichment for H3K4me3 and depleted H3K27me3. However, active promoters are marked by H3K4me3, and it could be a coincidence that NEAT1 also binds some of these promoters. At minimum, they should examine if H3K4me3 is reduced or transcription is reduced upon NEAT1 depletion. Is there any known interactions between NEAT1 and transcription factors or transcription machinery, which can support the activation role of NEAT1 for transcription?

Minor points:

1. In Fig 1H, they showed that histone H3 acetylation is increased on the NEAT1 locus upon p63 KD. They need to specify which acetylation marks were tested or specifically testing the ones that are known to associated with gene activation.

2. In p63 KD experiment, is HDAC1 binding to MALAT1 and NEAT1 loci reduced? They should measure this and demonstrate the correlation between reduced HDAC1 binding and increased NEAT1 expression.

3. In Fig. 4, they used ChIRP-seq to identify NEAT1 bound genomic regions. They should provide the mapping details for how many regions are mapped and show more detailed mapping results in supplemental data. More importantly, they should analyze whether there are any consensus motifs in those NEAT1 bound regions. Does NEAT1 bind to these regions through RNA:DNA interaction or through additional RNP?

4. They should provide a global view of how many gene promoters are bound by NEAT1 and how their expression is changed upon NEAT1 KD. Are most NEAT1 bound genes downregulated or only a few?

5. In the 3D culture (Fig. 5C), the basal layers appear to be abnormal. More careful studies should be done with additional basal markers such as Ecad, Krt5, basement membrane, proliferation and apoptosis markers etc.

Point-to-point response to the reviewer's comments

Novel data inserted

Main Figures

1. Figure 3G: IPA Upstream Regulator analysis of transcriptional nodes enriched in NEAT1-dependent transcriptional profile.
2. Figure 4B: NEAT1 ChIRP enrichment over Transcription Starting Site (TSS) and Transcription Termination Sites (TTS) based on the new ChIRP-seq in differentiated keratinocytes, in keeping with the previously reported ChIRP sequencing.
3. Figure 4C: New intersection of NEAT1 ChIRP enrichment over TSS against the indicated covalent histone modifications signatures in differentiated keratinocytes (H3K4me3, active genes, GSE98483; H3K27me3, transcriptionally silent genes, GSE175068).
4. Figure 4D: Barplot showing the top gene ontology (GO) terms for Biological Process of the intersection analysis between NEAT1 binding sites over TSSs and the RNA profile of differentiated keratinocytes (GSM1446880).
5. Figure 4E: Gene enrichment analysis of the indicated epidermal genes in the entire (ALL), top 50% and top 25% NEAT1 binding sites subgroups.
6. Figure 4F: new NEAT1 ChIRP enrichment and KLF4 genomic occupancy (GSE57702) over selected epidermal genes in differentiated keratinocytes.
7. Figure 4G: ChIP-qPCR showing the H3K4me3 epigenetic mark at the epidermal gene loci in differentiated keratinocytes upon NEAT1 depletion (LNA-NEAT1).
8. Figure 4H: Interactions between BRG1 and KLF4 with NEAT1 by RIP assay.
9. Figure 4I: ChIP-qPCR showing SFPQ occupancy at the NEAT1 binding site of the indicated epidermal genes loci.
10. Figure 5B: p63 mRNA levels by RT-qPCR in NEAT1 depleted keratinocytes
11. Figure 5E: percentage of Ki67 positive cells in control (NC) or NEAT1-depleted (LNA-NEAT1) organotypic human epidermis.
12. Figure 5F: confocal analysis of Laminin localization in control (NC) and NEAT1-depleted epidermis

Supplementary figures

1. Figure S1C: RT-qPCR analysis of NEAT1 RNA levels in control (SCR) and p63 depleted (sip63) cells
2. Figure S2B: p63 DNA binding sequence over *NEAT1* and *MALAT1* loci.
3. Figure S2E: ChIP-qPCR showing HDAC1 occupancy at MALAT1 and NEAT1 genomic loci in HEK293T cells transfected with scramble (SCR) or siRNA oligo targeting p63 (sip63).
4. Figure S2F: H3K27ac profile of the promoter region of MALAT1 and NEAT1 genomic loci in proliferating and differentiated keratinocytes.
5. Figure S3A: RNA FISH analysis of NEAT1₂ localization in differentiated keratinocytes.
6. Figure S3B: RT-qPCR analysis of the ratio of NEAT1₁/NEAT1₂ isoforms in differentiated keratinocytes.
7. Figure S4B: RNA FISH of endogenous NEAT1 and immunostaining of SFPQ in differentiated keratinocytes upon NEAT1 depletion.
8. Figure S6A and S6B: LacZ or antisense NEAT1 ChIRP enrichment over NEAT1-MALAT1 and SP3 loci in differentiated keratinocytes.
9. Figure S7A: genomic distribution of NEAT1 binding sites in differentiated keratinocytes.
10. Figure S7B: Barplot showing the top gene ontology (GO) terms for Biological Process of the intersection analysis between NEAT1 binding sites over TSSs and NEAT1-dependent RNA profile.
11. Figure S7C: qPCR analysis of the NEAT1 ChIRP enrichment over epidermal genes loci.

13. Figure S7D-E) new NEAT1 ChIRP enrichment and KLF4 genomic occupancy (GSE57702) over ZNF750 and KLKs genes in differentiated keratinocytes.
14. Figure S8: top three NEAT1 DNA consensus motifs enriched in the top 25% NEAT1 trans genomic sites.
15. Figure S9A: TUNEL assay in control (NC) and NEAT1-depleted (LNA-NEAT1) epidermis.
12. Figure S9B and S9C: TUNEL and cell cycle analysis in control (NC) and NEAT1-depleted (LNA-NEAT1) keratinocytes.
13. RNA-seq reads in control (NC) and NEAT1 depleted keratinocytes (LNA-NEAT1) together with NEAT1 ChIRP enrichment and KLF4 genomic occupancy (GSE57702) over the indicated epidermal genes in differentiated keratinocytes.
14. Figure S10: H&E staining (left panel) and quantification of stratum corneum thickness in NEAT1-depleted (LNA-NEAT1#2) organotypic human epidermis compared to control (NC) organotypic epidermis.
15. Figure S11: Immunofluorescence analysis of integrin $\alpha 6\beta 4$ localization in control (NC) and NEAT1-depleted (LNA-NEAT1) organotypic epidermis.

Due to space limitation, previous Figure 5B (RT-qPCR analysis of NEAT1 and MALAT1 RNA levels in differentiated keratinocytes upon NEAT1 or MALAT1 depletion) is now shown as Supplementary Figure S4A.

We replaced the GO term analysis of NEAT1 trans genomic sites over TSS genes (previous Figure 4E) with the GO term analyses of the intersection between NEAT1 binding genes and the RNA profile of differentiated keratinocytes (GSM1446880). As suggested by referee #1, this analysis is relevant to globally assess the functional link between NEAT1 binding over the genome and its function on epidermal differentiation.

The images of uncropped gels have been included in the source data file.

Point-to-point response to Reviewer #1

Reviewer: "The experiments are generally well executed and presented in clear and mostly compelling figures"

Response

We thank the reviewer for his/her positive comment.

Reviewer: "However, the connection between the LNA-NEAT1 RNA-seq and NEAT1 ChIRP results as presented is based on a few selected genes and currently provides limited insight into the general role of NEAT1 in epidermal biology. In addition, some of the other conclusions may be strengthened by providing additional controls (experimental and/or computational)."

Response

We completely agreed with this reviewer's comment. We tried to further improve the quality of the work by addressing all issues raised, as shown below. Hence, we hope that this revised version may be satisfactory for this referee.

Major points

1) *Reviewer: "The introduction lacks any mention of the known roles of lncRNAs in human epidermal biology. As the first experiment the authors present is a (targeted) expression profiling of lncRNAs after dNp63 knock-down, a section on what is currently known about lncRNA expression and function in human keratinocytes is warranted and should be included to provide context to the work presented in this manuscript."*

Response

We modified the introduction as requested; see the highlighted sentences on pages 3 and 4. Accordingly, we modified the discussion (see page 10), since we removed the part describing what is known about the link between lncRNA function and epidermal differentiation that was originally included in the discussion section.

2) Reviewer: *“As NEAT1 expression was not covered by the custom lncRNA array, it would be of interest, yet most of the work presented in the manuscript is on NEAT1 regulation and function, it would be of interest to add a figure on NEAT1 RT-qPCR after dNp63 knock-down in the 5 different cell lines (Figure S1). Especially as the justification for focussing on MALAT1 from this analysis is that it is upregulated after dNp63 knock-down in all 5 lines.”*

Response

We thank this reviewer for this comment. In new Figure S1C, we added the RT-qPCR analysis of NEAT1 and p63 RNA levels in A253, FaDu, hMEC and HCC1954 cells upon p63 silencing. p63 depletion increases NEAT1 RNA levels in all cell types. Please note that in Figure S1C we did not include the analysis performed in primary keratinocytes (HEKn) since it has been shown in Figure 1C.

3) Reviewer: *“The authors show predominant (potentially exclusive? Although this is difficult to assess from the provided images) localisation of NEAT1 in paraspeckles in differentiated keratinocytes. Do the authors think that its function and therefore the effect of NEAT1 LNA-mediated knock-down is associated with this localisation?”*

Response

We thank the reviewer for raising this issue. To test whether NEAT1 depletion resulted in paraspeckles disassembly, we performed FISH and immunofluorescence analysis in differentiated keratinocytes upon NEAT1 depletion. We found that LNA-mediated depletion of NEAT1 results in the disintegration of NEAT1- or SPFQ-associated paraspeckles (see new Figure S4B). This evidence indicates that NEAT1 depletion induces paraspeckles disassembly and this effect is associated with dysregulation of epidermal gene expression (see Figure 4). To further link paraspeckles localization and NEAT1 function to epidermal differentiation, we performed NEAT1 isoform specific FISH analysis with the super resolution software LEICA Lightning LasX (LEICA Stellaris 5). We found that in differentiated keratinocytes FISH probe detecting the 5' common region of NEAT1_1 and NEAT1_2 is exclusively detected in foci overlapping with those detected by NEAT1_2 specific probe (see new Figure S3A). This observation suggests that most of the NEAT1 signal detected in differentiated keratinocytes is associated with NEAT1_2 isoform, which is essential for paraspeckles assembly *in vitro* and *in vivo* (Clemson *et al*, 2009; Naganuma *et al*, 2012; Nakagawa *et al*, 2011; Yamazaki *et al*, 2021). In line with this evidence, our RNA-seq data and RT-qPCR analysis revealed that differentiated keratinocytes mostly express NEAT1_2 isoform (see new Figure S3B and response to point 4), referre#2). Furthermore, as described in the rebuttal to the next point, NEAT1 trans genomic sites are enriched for the DNA binding motif of the transcription factor ZKSCNA3 (see new Figure S8), which has been reported to be associated with the essential paraspeckles component ZNF24 (Fong KW *et al.*, 2013). Finally, in differentiated keratinocytes the paraspeckle protein SPFQ binds to epidermal gene promoters (see new Figure 4I) and NEAT1 binds to the epigenetic factor BRG1 (see new Figure 4H), an essential component of paraspeckles (Kawaguchi T. *et al.*, 2015) and a critical player of epidermal differentiation (Panatta E. *et al.*, 2020; Mardaryev AN *et al.*, 2014; Indra AK *et al.*, 2005; Bao X. *et al.*, 2013).

Collectively these data strongly indicate that the effect of NEAT1 on epidermal differentiation is likely to be linked to its paraspeckles localization. Of course, we cannot completely rule out the possibility that NEAT1_2 isoform might function outside the paraspeckles or that NEAT1_1 might cooperate with NEAT1_2 in controlling epidermal differentiation, based on its paraspeckles localization and its ability to increase the number of paraspeckles (Clemson *et al*, 2009; Naganuma *et al*, 2012). In conclusion, these evidences strongly suggest the involvement of NEAT1-associated paraspeckles in controlling epidermal differentiation.

4) Reviewer: *“In figure 4, the authors present ChIRP data to investigate NEAT1 genomic localisation and the potential consequences on transcription regulation of these regions. Does this suggest a model where NEAT1 ‘draws’ genomic regions into paraspeckles to regulate their expression? At the moment these observations seem unconnected and it is not clear how paraspeckle localisation of NEAT1 is important for keratinocyte biology beyond the observation that differentiated cells contain paraspeckles and proliferating cells do not and how it ties in with the role of NEAT1 on the genome”.*

Response

Paraspeckles are quite complex subnuclear structures formed by the lncRNA NEAT1 and multiple paraspeckles-associated proteins which include RNA binding proteins, splicing factors and transcription factors. NEAT1_2 long isoform is essential for the formation and maintenance of paraspeckles *in vitro* and *in vivo* as it acts as an architectural non-coding RNA promoting the assembly of these complex structures. As described in the previous point, we provide additional evidence suggesting that the effect of NEAT1 on epidermal differentiation is linked to its paraspeckles localization. Accordingly, we found that in differentiated keratinocytes NEAT1 depletion leads to paraspeckles disassembly (see new Figure S4B) and decrease of H3K4me3 epigenetic marker deposition on epidermal gene promoters (new Figure 4G), which ultimately results in epidermal gene expression dysregulation (Figure 4). Furthermore, we found that the paraspeckle’s protein SFPQ is localized over the NEAT1 transgenomic sites on selected epidermal genes (new Figure 4I).

Although not required for the assembly and maintenance of paraspeckle, the epidermal transcription factor KLF4 has been previously identified as a paraspeckles component by a mass spectrometry-based approach (Fong KW *et al.*, 2015). Remarkably, we found that in differentiated keratinocytes NEAT1 binds to the epidermal transcription factor KLF4 (see new Figure 4H) and NEAT1 profile is significantly enriched to genes whose expression is downmodulated upon silencing of KLF4 (see Figure 3G). Furthermore, by IPA Upstream Regulator analysis KLF4 has emerged as the top upstream transcriptional regulator that can explain the observed gene expression changes in our RNA-seq (see new Figure 3H). Collectively these data suggest that during epidermal differentiation NEAT1, by acting as an essential lncRNA of paraspeckles assembly, may facilitate the recruitment of critical epidermal transcription factors on differentiation genes promoting thus full activation of the differentiation program.

5) Reviewer: *“The authors present ChIRP data on the genomic localisation of NEAT1 in differentiated keratinocytes using a pool of 3 antisense capture oligos (COs). As a control they use the sense version of 1 of these COs. When interrogating the data (bigWig and called peak bed files downloaded from GEO using the provided reviewer token) it seems that ~45% of the 84,651 peaks called from the antisense probe ChIRP signals overlap with peaks (total of 120,429 peaks) called from the antisense (control) probe signals. Moreover, in the genome-browser (viewing the bigwig files) when there is signal in the antisense track, there usually is signal in the sense (control track), albeit weaker as indicated by the authors in the manuscript (supplemental figure S6), which does overlap with H3K4me3 ChIP-seq signal.*

Can it be excluded that these differences in signal may have arisen from the fact that the authors used 3 antisense COs and 1 sense CO?

Response

These are valuable considerations, and we thank the reviewer to raise these issues. We agree with this referee that there is a significant overlap between the CHIRP signal generated with the three antisense oligos (CO) and those generated by a single sense oligo. As suggested by this referee we performed ChIRP-qPCR analysis on few NEAT trans sites with orthogonal set of CO (see Figure 1 below). Although we observed an enrichment with the antisense oligos we also detected a consistent background generated with the sense oligos.

Figure 1. A) NEAT1 levels upon RNA pulldown utilizing three oligos antisense (CO1, CO2, and CO3) and the relative oligos sense as control. B) ChIRP-qPCR validation of NEAT1 enrichment on the indicated genomic loci.

The choice of the sense oligos as control for ChIRP is based on previous publication (West et al., 2014). However, it is possible the sense oligos do not represent reliable controls since it has been demonstrated that NEAT1 is able by itself to bind to DNA (Senturk Cetin et al., 2019). Along NEAT1 sequence there are significant triplex-forming regions (TFR) with the ability to interact with specific DNA sequence (Senturk Cetin et al., 2019). Our sense oligos are localized in the first 150 nucleotides of NEAT1 sequence in correspondence of a significant TFR (see Figure 5H of Senturk Cetin et al., 2019). It is noteworthy that additional ChIRP studies (Chakravarty D. et al., 2014; Wen S. et al., 2020; Chu C. et al., 2012) as well as commercially available NEAT1 ChIRP assay (Magna ChIRP™ NEAT1 lncRNA Probe Set #03308, by Millipore) utilize LacZ as control for ChIRP-seq approach. Therefore, we also validated our data by utilizing three oligos LacZ as control. As shown in the new Figure S7C, ChIRP-qPCR confirmed NEAT1 enrichment on NEAT1 promoter as well as on KLK5, DLX5 and DSC2 epidermal gene promoters. To confirm these results at global level, we performed a new ChIRP-seq utilizing LacZ as control. As shown in new Figure 4 and S7 we were able to confirm: *i*) NEAT1 ChIRP enrichment over Transcription Starting Site (TSS) (panel 4B); *ii*) enrichment of NEAT1 TSSs targets for the active chromatin-associated histone H3 modification H3K4me3 (panel 4C); *iii*) NEAT1 binding to genes critically involved in epidermal differentiation and integrity (panels 4F and Figure S7). Based on these results, we decided to draw our conclusions on the basis of this new ChIRP-seq data. Accordingly, we deposited the new ChIRP-seq raw data in the NCBI's Gene expression Omnibus (GSE205960).

6) Reviewer: "Can the authors exclude a contribution of non-specific binding of the probes to actively transcribed open chromatin regions?"

Response

To exclude a contribution of non-specific binding of the probes to actively transcribed open chromatin regions, we tested whether NEAT1 depletion affects the H3K4me3 marks on epidermal genes. To this aim we performed H3K4me3 ChIP-qPCR and we found that NEAT1 silencing decreases H3K4me3 marks on the promoters of NEAT1 bound genes KLK5 and DLX5 (see new Figure 4G).

7) Reviewer: "Are the NEAT1 binding regions enriched in specific DNA sequence motifs and how do they relate to the COs used in the ChIRP assay?"

Response

To address this issue, we performed an analysis on top 25% NEAT1 trans genomic sites and we identified several NEAT1 DNA consensus motifs (see new Figure S8). Remarkably, the DNA binding motif of the transcription factor ZKSCNA3 is one of the top DNA binding motifs enriched in NEAT1 trans genomic sites. Since ZKSCNA3 has been reported to be associated with the essential paraspeckles component ZNF24 (Fong KW et al., 2013), this data suggests that NEAT1 binding to chromatin is likely to occur in association with paraspeckles proteins. Accordingly, we found that the paraspeckle protein SPFQ is localized to the NEAT1 bound epidermal gene promoters (see new Figure 4I).

Regarding the relation between the identified DNA binding motifs and the COs used in the ChIRP assay, we performed a motif comparison of the oligonucleotides against all possible Transcription factors binding motif using TomTom (Gupta et al., 2007) and we did not find any significant correlation.

8) Reviewer: *“The authors may need to include more substantial controls to convince the eventual readers of the strength of the ChIRP dataset and the conclusions drawn based thereon. For instance, ChIRP after LNA-NEAT1 transfection, ChIRP with an orthogonal set of COs. These may be done using sequencing as a read-out, but potentially a qPCR read-out for representative binding sites may already suffice. Alternatively, the authors can choose to explicitly state the limitations of the dataset as generated and presented currently in the manuscript text and discuss the impact on the strength on the conclusions that can be drawn at this stage.”*

Response

As described in the previous points, we decided to improve the limitations of our previous ChIRP-seq data by performing an additional ChIRP-seq experiment and including substantial controls. We really hope that these new data may improve the robustness and strength of our conclusions.

9) Reviewer: *“The authors show an overlap between NEAT1 ChIRP signals and a small selected set of NEAT1 dependent differentiation associated genes (Figure 4F). It is unclear from the presented analyses whether this proposed regulation of differentiation genes by NEAT1 binding is a general principle or not. As the authors have a complete set of NEAT1 responsive genes (Figure 3) as well as genome-wide NEAT1 binding data, it would be of interest to include a more global analysis of the claim that NEAT1 controls differentiation genes. Some specific questions relevant to this are: What globally happens to the expression of NEAT1 bound genes during differentiation and upon LNA-NEAT1 silencing? What are the proportions of genes whose expression goes up/down. Are these enriched in genes involved in specific processes (GO term analysis)?”*

Response

We thank the reviewer to raise these questions. We performed an intersection analysis between the NEAT1 bound genes and the RNA profile of differentiated keratinocytes (GSM1446880). We found that NEAT1 binds the promoter of 651 and 116 genes whose expression is induced and repressed during keratinocytes differentiation, respectively. GO term analyses revealed that the 651 genes categorize in GO terms associated with regulation of transcription and cell differentiation (average FDR enrichment $10e^{-10}$) (see new Figure 4D). GO term analysis of the 116 genes produce low FDR enrichment ($10e^{-2}$), mostly in cell cycle related pathways. This data indicates that NEAT1 is preferentially found on those epidermal genes whose transcription is induced upon the activation of the epidermal differentiation program. We also crossed our NEAT1 gene profile with the NEAT1 binding data. Although we found that NEAT1 binds NEAT1 activated or repressed genes with similar extent (approximately 41 genes), the GO term analyses revealed that the subgroup of genes bound and activated by NEAT1 categorize in GO terms associated with epidermis development, establishment of skin barrier and skin development (see

new Figure S7B). Based on these data, we can argue that NEAT1 binding to the promoter regions does not imply per se transcriptional activation and it is likely that the NEAT1 effect on specific subset of gene promoters may be dictated by its interaction with epidermal factors (for instance KLF4) or by other chromatin determinants. We discussed this point on page 14.

10) Reviewer: “What about the overlap of these NEAT1 bound genes with Δ Np63 (repressed) target genes?”

Response

Δ Np63 repressed genes have been mainly validated and characterized in proliferating keratinocytes or in tumor cells. Conversely, our ChIRP-seq analysis have been performed in differentiated keratinocytes, which do not express Δ Np63 (see Figure 2A). One possibility is that the referee asked us to analyze the overlap between NEAT1 bound genes and those Δ Np63 repressed target genes whose expression is increased during epidermal differentiation. The subgroup of validated Δ Np63 repressed genes include a limited number of genes. Furthermore, based on GSE18590 dataset, the expression of few Δ Np63 repressed genes (HES1, KLF4, HYAL1 and IGFBP3) increases during epidermal differentiation (see Figure 2 below). Therefore, the evidence that Δ Np63 represses the transcription of specific subset of genes does not imply that these genes are modulated during differentiation. Nevertheless, among HES1, KLF4, HYAL1 and IGFBP3 genes, KLF4 is the only Δ Np63 repressed gene which is represented in the top 25% NEAT1 bound genes.

Figure 2. The expression of the indicated Δ Np63 repressed genes have been analyzed in primary human keratinocytes in high calcium differentiation conditions (Gene Expression Omnibus GSE18590) (Sen et al 2010).

Although this evidence strengthens our idea that KLF4 may be functionally important for ability of NEAT1 to regulate epidermal differentiation (see previous point #4, point 4 of reviewer #2 and point 5 reviewer #3), we believe that this analysis is not so informative for the limited number of genes analyzed. For this reason, we decided to do not include this analysis in the paper. Of

course, if we did not correctly understand this point, we will be happy to revise the manuscript accordingly.

11) Reviewer: “The 3D organotypic human epidermal equivalent experiments are non-trivial and well executed. The effect on epidermis thickness is convincing. Moreover, it seems that not only the thickness of the cornified layer is affected, but also that the granular layer is absent (based on morphology and lack of the typical granulated nuclei in the H&E stained sections in Figure 5C). There does seem to be some discrepancy between the LNA-NEAT1#1 and LNA-NEAT1#2 in terms of ZNF750 expression. Have the authors confirmed the effect on cornified layer thickness using the second LNA as well?”

Response

We confirmed that NEAT1 depletion by LNA-NEAT1#2 leads to reduction of the stratum corneum thickness (see new Figure S10).

12) Reviewer: “The increased and decreased level of NEAT1 expression in Ichthyosis and Psoriasis, respectively, may reflect changes in ratios of non-differentiated versus differentiated cell populations in these diseases, rather than a specific association of NEAT1 with these afflictions. For instance, do other (classical) markers of differentiation show a same/similar association?”

Response

We analyzed the expression of several markers of epidermal differentiation in human lamellar Ichthyosis, in skin samples of ALOX12B knock-out mice, a mouse model resembling congenital Ichthyosis (Krieg et al, 2020), and in psoriatic lesions. As shown in Figure 3, there is not a general association between expression of differentiation markers (FLG, IVL, KRT10, KRT16, TGM1) and ratios of non-differentiated versus differentiated cells we expected in these skin diseases.

Figure 3. A) Analysis of the expression of the indicated epidermal differentiation markers in psoriasis (GSE13555) dataset. Expression value is shown as the mean \pm SD of 64 (n=64, normal skin) and 58 (n=58, psoriatic lesions) samples. p value was calculated using Student’s t test. B) Analysis of the expression of the indicated epidermal differentiation markers in human biopsies of lamellar Ichthyosis (GSE108640) and in Alox12b Knock-out (KO) (GSE127434). In lamellar Ichthyosis dataset, the expression value is shown as the mean \pm SD of 14 (n=14, normal skin) and 6 (n=6 lamellar Ichthyosis lesions) samples. In Alox12b KO dataset the expression value is shown as the mean \pm SD of 4 (n=4, normal skin) and 5 (n=5 Ichthyosis lesions) samples. p value was calculated using Student’s t test.

The expression of KRT10, TGM1, IVL and KRT16 is not decreased in high-proliferative psoriatic lesions (Figure 3A). Similarly, in mouse and human Ichthyosis samples we did not observe an increased expression of IVL and KRT10 (Figure 3B). These results suggest that NEAT1 dysregulation might not simply reflect changes in ratios of non-differentiated versus differentiated cell populations in these skin diseases. However, we are aware that our study is far from formally proving that NEAT1 dysregulation may have a role on the pathogenesis of these complex skin diseases. Ichthyosis and psoriasis are complex skin diseases characterized not only by alteration of the proliferation/differentiation balance but also by immune dysregulation. Based on these considerations, we down-toned our conclusions on the potential link between NEAT1 dysregulation and the pathogenesis of these skin diseases (see modified text in the Abstract and Discussion, page 14)

Minor points:

i) Reviewer: *“As written now, the title grammatically seems to suggest that the regulation of NEAT1 expression by dNp63 takes place during differentiation. However, the authors convincingly show that the regulation takes place, in the form of HDAC-mediated repression, in proliferating (non-differentiated) cells. Therefore, the title does not seem to represent the conclusions in the most intuitive way”.*

Response

We thank the reviewer for this valuable consideration. We agree the title may be misleading and therefore, we decided to change it. The new title is “The long non-coding RNA NEAT1 is a Δ Np63 target gene modulating epidermal differentiation”

ii) Reviewer: *“In the Methods section, the description of the differentiation induction does not include at which level of confluency of the culture the CaCl₂ was added. This is a key parameter in these assays and should be included.”*

Response

We included this information in the Methods section (see page 13).

iii) Reviewer: *“What was the knock-down efficiency of NEAT1 and MALAT1 in the samples used for RNA-seq analysis described in Figure 3. Are these the same samples as depicted in Figure 5A? If so, please state. If not, please indicated knock-down efficiency.”*

Response

We thank the reviewer for this question. We silenced NEAT1 or MALAT1 in differentiated keratinocytes and then we extracted both RNA, which we used for RNA-seq (Figure 3), and proteins lysates we used of IB analysis (Figure 5A). To improve the clarity of the manuscript, we decided to move the panel showing the knock-down efficiency (Figure 5B) in Figure S4A. We referred to panel S5A when we described the impact of NEAT1 depletion on the expression of the differentiation markers (see page 9).

iv) Reviewer: *“In figure 4 the authors show overlap between NEAT1 binding regions and H3K4me3 marked transcription start sites, suggesting that these NEAT1 bound genes are actively transcribed. Did the authors extend these analyses to investigate a quantitative relationship between NEAT1 signal and gene expression level (eg steady state RNA abundance by RNA-seq)?”*

Response

Gene expression levels depend on many factors and NEAT1 bound on genes promoters could not be the only determinant for gene expression level. Furthermore, a quantitative relationship between

NEAT1 signal and gene expression levels might be misleading since it is based on the comparison of two datasets generated by two completely different experimental approaches (RNA-seq vs ChIRP-seq). Therefore, we respectfully believe that such analysis might produce not reliable conclusions.

Point-to-point response to Reviewer #2

Reviewer: *“This study nicely describes the novel observation that the NEAT1 lncRNA controls epidermal differentiation. The manuscript is thoughtful, clear and well written with logical flow and clear description of rational and experiments.”*

Response

We thank the reviewer for this positive comment.

1) Reviewer: *“The study begins with a lncRNA library screen to identify RNAs that are regulated by the epidermal transcription factor deltaNp63. However, NEAT1 is not included in the library, so does not factor in the screen. The authors focus on a different RNA MALAT1, that is regulated by p63 but ultimately has no role in controlling differentiation. The link between this work and the authors switch to working on NEAT1 is unclear. Why was it not included in the initial screen, and how was it chosen for follow up without that data?”*

Response

lncRNA Expression profiling analysis was performed using an in-house microarray platform constructed by our co-author and collaborator Prof. George A. Calin at The University of Texas MD Anderson Cancer Center. This in-house microarray platform does not include the probe specific for the long-non coding NEAT1. Conversely, MALAT1 (also known as NEAT2) was included in this lncRNAs custom-designed microarray. We decided to include in our investigation the lncRNA NEAT1 since several observations suggest that MALAT1 and NEAT1 might be functionally interconnected. In detail, these observations are: i) the analysis of the 3D chromatin interactions in human cells which localizes MALAT1 genomic locus in close proximity to the NEAT1 locus (Jin et al, 2013) ii) the adjacent nuclear localization of speckles (MALAT1 positive foci) and paraspeckles (NEAT1 positive foci) (Fox et al, 2002); iii) the ability of NEAT1 and MALAT1 to co-regulate a subset of common genes (West et al, 2014).

As requested by referee #1, we have further validated the ability of Δ Np63 to repress NEAT1 expression in A253, FaDu, hMEC and HCC1954 cells (see new Figure S1C). These cells have been utilized for our initial microarray-based screening (see Figure 1A).

2) Reviewer *“Regardless, the authors show comprehensively that NEAT1 expression is repressed by deltaNp63 mediated recruitment of HDACs, and that concomitant with differentiation and loss of deltaNp63, NEAT1 levels increase. They show conclusively that NEAT1 plays a role in differentiation and induction of epidermal gene expression, by localization to target gene promoters, and loss of their expression upon NEAT1 knockdown. Finally, the authors propose a role for NEAT1 activity in epidermal disease, as an increase in NEAT1 expression is seen in the hyper differentiation disease Ichthyosis. This link is more tenuous and the authors do not show data describing overexpression of NEAT1 in their models system. If they wish to make this claim they should show data describing the overexpression phenotype. If NEAT1 is overexpressed in keratinocytes can it drive differentiation?”*

Response

We agree with this referee that the functional link between dysregulation of NEAT1 expression and the pathogenesis of skin diseases is rather preliminary. Psoriasis and ichthyosis are quite

complex skin diseases and alterations of the proliferation/differentiation balance together with immune dysregulation are the main determinants of these skin diseases. As suggested by the referee, gain of function studies might be indicative on the potential role of NEAT1 in driving epidermal differentiation and strengthen the link between NEAT1 and skin diseases. These approaches are quite challenging in primary or immortalized keratinocytes since plasmid transfection efficiency is too low (see Figure 4A below) to detect global changes of differentiation markers expression. Furthermore, lentiviral infection is challenging, if not possible, since NEAT1_2 isoform, which is the most expressed NEAT1 isoform in differentiated keratinocytes and an essential component of paraspeckles (see referee #1, points 3 and 4; referee #3 point x), is ~23 kb and cannot be packaged in lentiviral particles.

Figure 4. A) Human primary keratinocytes (HEKn) or human immortalized keratinocytes (Ker-CT) were transfected with GFP expressing vector. GFP was visualized 24 hours after transfection. B) Ker-CT cells were electroporated with the indicated PMO oligos. Forty-eight hours after electroporation NEAT1 long isoform (NEAT1_2) RNA levels were

quantified by RT-qPCR. C) Ker-CT treated as in B) were shifted in differentiation medium (high calcium) for 24 hours and the expression of the indicated differentiation markers was analyzed by RT-qPCR. Data shown are the mean of two biological replicates \pm SD

To circumvent these challenges and address reviewer's suggestion, we decided to exploit an alternative approach aimed to increase endogenous NEAT1_2 RNA level. We utilized a targeted antisense oligonucleotide (ASO) approach to sterically block NEAT1_1 polyadenylation processing, achieving thus specific upregulation of NEAT1_2 RNA levels. This approach has been successfully utilized by Archa Fox's group to demonstrate that increasing endogenous NEAT1_2 drives paraspeckles assembly and this effect is associated with increased differentiation of high-risk neuroblastoma cells (Naveed A. et al., 2020). Technically, we utilized morpholino oligos (PMO) spanning the sequence -14 to +5 (PMO_PSA_NEAT1_1) respect the polyadenylation site (PSA) of NEAT1_1 RNA (Naveed A. et al., 2020). Immortalized keratinocytes Ker-CT were electroporated with Necleofector using control (PMO_CTRL) or PSA targeting oligos (PMO_PSA_NEAT1_1). We firstly validated the effect of PMO_PSA_NEAT1_1 oligo on NEAT1_2 RNA levels. As shown in Figure 4B, PMO_PSA_NEAT1_1 oligo increases NEAT1_2 levels in keratinocytes. Then, we analysed the expression of few epidermal differentiation markers in keratinocytes at early time point after high calcium treatment. We choose those epidermal differentiation genes which are bound by NEAT1 and whose expression is impaired upon NEAT1 depletion. We found that upregulation of NEAT1_2 expression is associated with the increase of KLK5, KLK6, DLX5, DSC2 and ZNF750 RNA levels (Figure 4C). These data suggest that NEAT1_2 may facilitate or accelerate the activation of the differentiation program, at least in our experimental conditions. However, we believe that these data, although supporting our model, do not prove that NEAT1_2 might be sufficient per se to drive the full activation of the epidermal differentiation program. The epidermal differentiation program is the result of intricate signalling pathways and transcriptional nodes and it is more reasonable that NEAT1 might participate in this intricate network contributing to the full transcriptional activation of the epidermal differentiation genes. Based on these considerations, we decided not to include these data in the manuscript and down tone our conclusions related to the link between NEAT1, aberrant differentiation and Ichthyosis (see page 9 and 12).

In case that this referee believe that these data might be informative for the scientific community, we will be happy to include in a Supplementary Figure.

3) Reviewer *“Similarly, is there a feedback loop between NEAT1 and deltaNp53? Can NEAT1 expression repress deltaNp53 expression?”*

Response

We analyzed the expression levels of Δ Np63 in NEAT1 depleted keratinocytes. We did not observe any significant changes of Δ Np63 RNA levels upon NEAT1 silencing (see new Figure 5B). Accordingly, Tp63 gene is not included in the subgroup of genes affected by NEAT1 depletion in our RNA-seq analysis (see Figure 4).

4) Reviewer *“Finally is it not clear how NEAT1 is impacting gene expression. The authors show its localization to the TSS of epidermal genes, but what is the proposed model for its activity there? Is it acting as a scaffold for RNA binding proteins that promote transcription?”*

Response

As described in point 3 and 4 of referee #1 response, our additional data strongly suggest that NEAT1 function on epidermal differentiation is functionally related to its ability to act as a critical component of the paraspeckles (see new Figures S4B). Many paraspeckle proteins are RNA-binding proteins, such as NONO and SFPQ, RBM14, EWSR1, FUS, TAF15 and TDP-43, or heterogeneous nuclear ribonucleoproteins (HNRNP), such as (HNRNP)K, HNRNPA1, HNRNPR, HNRNPUL1 (see review by Tomohiro Yamazaki and Tetsuro Hirose). These RNA binding proteins act as multifunctional paraspeckle proteins that are involved in multiple gene expression processes, including transcriptional regulation, pre-mRNA splicing, mRNA stability, and translation. In addition to RNA binding proteins, paraspeckle contains multiple transcription factors. Although not required for the assembly and maintenance of paraspeckle, the epidermal transcription factor KLF4 has been previously identified as a paraspeckles component by a mass spectrometry-based approach (Fong KW et al., 2015). KLF4 transcriptional activity is critical for the proper transcriptional activation of epidermal differentiation genes (Bao X. et al., 2013; Boxer LD. Et al., 2014). Remarkably, NEAT1 profile is significantly enriched in genes whose expression is downmodulated upon KLF4 silencing (see Figure 3X) and IPA Upstream Regulator analysis unveiled KLF4 as top upstream transcriptional regulator that can explain the observed gene expression changes in our RNA-seq (see new Figure 3H). These data prompted us to validate NEAT1/KLF4 binding in differentiated keratinocytes. To this aim we performed RNA immunoprecipitation (RIP) assay and we found that in differentiated keratinocytes KLF4 interacts with NEAT1 (see new Figure 4X). Furthermore, KLF4 DNA binding regions in DLX5, ZNF750, EGR3 epidermal genes partially overlaps or is localized in proximity of the NEAT1 trans genomic sites (see new Figure 4H). As control of the RIP assay, we utilized BRG1 (also known as SMARD4), one of the catalytic ATPase subunit SWI/SNF chromatin remodeling complex (Wu et al., 2009). BRG1 interacts with NEAT1 and is essential for the assembly and maintenance of these structure (Kawaguchi T. et al., 2015). Remarkably, BRG1 is also critically involved in the regulation of epidermal differentiation genes and cooperates with KLF4 to induce the expression of epidermal differentiation genes (Bao et al., 2013). Collectively, these data suggest a model in which NEAT1 facilitates the recruitment of critical epidermal transcription factors on differentiation gene promoters promoting thus the activation of the differentiation program.

5) Reviewer: *“This work is a solid contribution to the field and provides a novel component to the control of epidermal differentiation. As such I approve its publication in nature communication after the issues described above are addressed.”*

Response

We thank the reviewer for this positive comment. We tried to further improve the quality of the work by addressing all issues raised. Hence, we hope that this revised version may be satisfactory for this referee.

Point-to-point response to Reviewer #3

Reviewer: *“In this manuscript, Fierro et al., studied the regulation of NEAT1, a well-known long-noncoding RNA, by dNp63, and the function of NEAT1 in the differentiation of human keratinocytes. The major findings are the repression of NEAT1 by dNp63 through the recruitment of HDAC in proliferative keratinocytes, the function of NEAT1 in promoting epidermal differentiation, and the direct binding of NEAT1 on the promoters of several key regulation of epidermal differentiation, including ZNF750, KLF4 and DLX5. Overall, this study is generally well done and provides interesting insights into the regulation and function of NEAT1 in human epidermal differentiation.”*

Response

We thank the reviewer for this positive comment.

Major points:

1) Reviewer: *“ChIP assays, including both ChIP-seq and ChIP-PCR, could suffer from non-specific crosslink of TF target to DNA sequences, in particular the ones close to the TSS. To firmly establish the direct binding of p63 to the binding sites on MALAT1 and NEAT1 loci, they should identify the canonical p63 motif within the peak”*

Response

In Figure S2B we now show the p63 DNA binding motif in *NEAT1* and *MALAT1* loci.

2) Reviewer: *“In addition, it is intriguing that p63 negatively regulates MALAT1 and NEAT1 through the recruitment of HDAC to these sites. It will be interesting to identify the determinant for the activation vs repression function of p63 e.g. in which context p63 recruits HDAC and in which context p63 recruits pol II? If they clone the binding site and perform promoter/enhancer assays in the same cells, will they still observe negative regulation? These additional studies can strengthen the proposed mechanism mediated by p63”.*

Response

The outcome of Δ Np63 transcriptional activity likely relies on its ability to bind to specific chromatin remodelling or epigenetic factors. For instance, Δ Np63 binds to the methyltransferase KMTD2 in order to transcriptionally activate epithelial genes (Lin-Shiao et al., 2018), while it recruits the SWI/SNF subunit ACTL6A to induce the repression of specific target genes (Panatta et al., 2020; Bao et al., 2013). It has been previously demonstrated that HDAC1/2 act redundantly to mediate repressive functions of p63 in epidermal progenitor cells (LeBoeuf M et al., 2010). Here, we provided evidence indicating that Δ Np63 exploits an HDAC-dependent mechanism to repress NEAT1 and MALAT1 transcription. As suggested by the referee, promoter/enhancer assays could strengthen this conclusion. To this aim, we cloned the sequence (200 bp) spanning the p63 DNA binding motif of NEAT1 and MALAT1 promoters into the pGL3 basic and promoter vector. We adopted two complementary approaches. We first tested the effect of exogenous Δ Np63 on NEAT1 and MALAT1 promoters in H1299 cells (Δ Np63 negative). We did not observe any significant effect of Δ Np63 on the luciferase activity driven by the p63 binding site of NEAT1 and MALAT1 promoter (Figure 5A). Conversely, exogenous Δ Np63 markedly induces the luciferase activity of pGL3 basic-KRT14 promoter (KRT14 is a known Δ Np63 transcriptional

target gene whose expression is physiologically induced by Δ Np63) (Figure 5A). As alternative approach, we tested whether p63 silencing affects KRT14, NEAT1 or MALAT1 promoter activity. We did not observe any significant effect of p63 depletion on luciferase activity driven by NEAT1 or MALAT1 promoter (Figure 5B). Conversely, KRT14-driven luciferase activity decreases upon p63 depletion. These data indicate that the p63 binding sites of NEAT1 and MALAT1 promoters, although acting differently respect to a canonical Δ Np63 activated gene, are not sufficient to determine Δ Np63-dependent repression. Although circular DNA can be potentially assembled in nucleosome, it is possible that circular DNA-nucleosome complex does not entirely recapitulate the chromatin architecture of NEAT1 or MALAT1 promoters in the genome. Additionally, it is also possible that the region we cloned is not sufficient to act as determinant of Δ Np63-mediated repression. As described in the next point, we provided additional evidence, such as the p63-dependent recruitment of HDAC1 on NEAT1 promoter (see new Figure S2E), strengthening the involvement of HDAC in the Δ Np63-mediated repression of NEAT1 and MALAT1 transcription.

Figure 5. A) H1299 cells (Δ Np63 negative) were transfected with the indicated pGL3 luciferase gene construct holding p63 binding site (~220 bp) of NEAT1 or MALAT1 human promoter together with either an empty vector (EV) or with the Δ Np63 expressing vector. Co-transfection of a renilla luciferase control plasmid was used to normalize the transfection efficiency. Luciferase assay was performed 24 hrs after transfection. Data are shown as the mean \pm SD of three replicates. Whole cell extracts were used to verify the expression levels of exogenous Δ Np63 by IB. B) Human immortalized keratinocytes were transfected with scramble oligos (SCR) or oligos targeting Tp63 mRNA (sip63). 24 hrs after sip63 transfection, cells were transfected with the indicated pGL3 luciferase gene construct. Co-transfection of a renilla luciferase control plasmid was used to normalize the transfection efficiency. Luciferase assay was performed 24 hrs after the last transfection. Data are shown as the mean \pm SD of three replicates. Whole cell extracts were used to verify the expression levels of endogenous Δ Np63 by IB.

3) Reviewer: “Previous studies (Standaert et al., RNA 2014; Adriaens et al., RNA 2019) have shown that genetic KO of NEAT1 (both NEAT1 and NEAT1_2) causes reduced proliferation during mammary gland development and NEAT1 (the short isoform) is dispensable for the function. Although it is possible that mouse and human NEAT1 function may be different, they should carefully address the potential differences. At minimum, they should check whether NEAT1 KD by LNA can alter cell proliferation in raft culture experiments in Fig. 5. And they should discuss how mouse NEAT1 appears to be required for cell proliferation in mammary gland whereas human NEAT1 appears to promote epidermal differentiation.”

Response

We performed Ki67 staining in control and NEAT1 depleted organotypic human epidermal model. We did not observe any significant changes in the percentage of Ki67 positive cells (see new Figure 5E). To further corroborate this result, we tested whether NEAT1 depletion impact keratinocytes proliferation in 2D cell culture. As shown in new Figure S9B NEAT1 depletion does not affect cell cycle phase distribution in proliferating as well as differentiated cells. We have analyzed the effect of NEAT1 depletion at early time point upon differentiation (24 hrs in differentiation medium) since most of our experiments have been performed in NEAT1-depleted differentiated keratinocytes. We did not observe any alteration of cell survival in NEAT1-depleted keratinocytes and in NEAT1-depleted epidermis (new Figure S9A and S9C). These data indicate that in human primary keratinocytes NEAT1 depletion does not impact neither cell proliferation nor cell survival, at least in our experimental model.

As promptly pointed out by the referee, previous reports demonstrated that during mammary gland development NEAT1 KO alveolar cells show reduced proliferation rate respect to wild-type cells. However, this phenotype is evident in alveolar cells at midgestation (8.5 and 12.5 d post-coitum), but not at the start of pregnancy (Standaert et al., RNA 2014), suggesting that cytostatic effect exerted by NEAT1 genetic deletion is likely to be context dependent. Primary keratinocytes and alveolar cells could express different levels of NEAT1 RNA and respond differently to NEAT1 depletion. In addition, the discrepancies of the phenotypes observed between human primary keratinocytes and murine alveolar cells could be also related to difference of NEAT1 depletion efficiency (KO vs silencing). Furthermore, difference in mouse and human NEAT1 function and stability may also explain this discrepancy. Indeed, mouse NEAT1_1 and NEAT1_2 isoforms are highly unstable lncRNA, while human NEAT1 is relatively stable (Clark MB. Et al., 2012). Since many paraspeckles proteins (e.g NONO, SFPQ) contribute to NEAT1 RNA stabilization (Yamazaki and Hirose 2015), it is possible that the dynamic of paraspeckles formation and maintenance is different in human and mouse cells. Consequently, NEAT1 depletion could exert different outcomes in human and mouse cells.

It is noteworthy that the link between NEAT1 and cellular differentiation has been described in additional cellular contexts. For instance, NEAT1 expression is upregulated during differentiation of neurons, glia, myeloid cells and muscle, although the molecular details of its differentiation-mediated regulation have not been elucidated (Mercer *et al*, 2010; Sunwoo *et al*, 2009; Zeng *et al*, 2014).

Notably, the functional link between NEAT1 function and cellular differentiation has been also postulated in pathological context. In pancreatic cancer NEAT1 acts as tumor suppressor by regulating the expression of pancreatic differentiation genes (Mello & Attardi, 2018; Mello *et al.*, 2017). More recently, in neuroblastoma the upregulation of NEAT1_2 isoform by morpholino oligo targeting the polyadenylation sites is associated with increased expression of differentiation genes (Naveed A. et al., 2021).

4) Reviewer: *“They used ChIRP-seq to identify NEAT1 associated DNA sequences and identified the binding of NEAT1 to the promoters of several important differentiation genes, such as ZNF750, KLF4 and DLX5. They should distinguish whether such binding is mediated by NEAT1 (short isoform) or NEAT1_2 (long isoform). To do so, they should first distinguish the ration between the short and long isoforms, based on RNA-seq data. If there are significant portions of short and long isoforms, they may need to revise their ChIRP-seq approach since all 3 probes bind to the shared 5’ regions of both short and long isoforms.”*

Response

As shown in Figure 6A below, our RNA-seq data indicate that in differentiated keratinocytes NEAT1_2 isoform (~23kb) is expressed at higher level respect to NEAT1_1 short isoform. However, it is possible that this RNA-seq-based data underestimates the amount of NEAT1_2

since it has been demonstrated that classic RNA extraction procedures (Trizol or RNAeasy from Qiagen) do not allow the complete extraction of the NEAT1 long isoform (Chujo T. et al., 2017). Chujo and colleagues demonstrated that heating cell lysate (65° for 10 minutes) in RNA extraction reagent markedly improves NEAT1_2 extraction. As shown in Figure 6B below, we validated this improved RNA extraction method using lysates from differentiated keratinocytes.

Figure 6. A) RNA-seq based TPM (Transcripts Per Kilobase Million) of the indicated NEAT1 isoforms. B) Differentiated keratinocytes were lysed in RNA extraction reagent (RNAeasy from Qiagen) and then treated at 65° for 10 minutes. Total RNA was then purified and NEAT1_2 RNA levels were quantified by RT-qPCR analysis. C) RT-qPCR analysis of NEAT1_1 and NEAT1_2 ratio in differentiated keratinocytes.

Based on this result, we decided to utilize this method to measure the ratio between the short and long isoform by RT-qPCR. To do this, we normalized the PCR amplification by calculating primers efficiency (see Materials and Methods section). As shown in new Figure S3C, in differentiated keratinocytes NEAT1_2 isoform, which acts as an essential architectural RNA for paraspeckles assembly represents almost the 80% of total NEAT1. Accordingly, we found that NEAT1 function on epidermal differentiation is associated with its paraspeckles localization (see new Figure S4B). As correctly stated by the referee, our ChIRP-seq approach exploited three antisense biotinylated probes complementary to the sharing 5' region of both short and long NEAT1 isoforms. Performing ChIRP-seq with NEAT1_2 specific probe could be challenging since NEAT1_2 isoform is associated with multiple RNA binding proteins and transcription factors in the core of paraspeckles. Conversely the 5' region of NEAT1_2, similarly to the short NEAT1_1 isoform, is localized in the shell of paraspeckles and it is likely more accessible for RNA pulldown. In line with this, ChIRP-seq data published so far utilized NEAT1 probes directed against the 5' RNA region (Chakravarty D. et al., 2014; Wen S. et al., 2020; Chu C. et al., 2012). Furthermore, it is possible that NEAT1_1 short isoform, being localized into paraspeckles and facilitating paraspeckles assembly (Clemson *et al*, 2009; Naganuma *et al*, 2012) might contribute to NEAT1 binding on trans genomic sites. Although we can not formally prove the contribution of each NEAT1 isoforms on ChIRP seq signal at this stage, we believe that our ChIRP-seq data are nevertheless informative on the function of NEAT1-associated paraspeckles on epidermal differentiation.

5) Reviewer: “Fundamentally, they need to probe deeper for the mechanism of how the binding of NEAT1 on the promoter of these genes promotes their expression. Their current data only show that NEAT1 bound promoters have enrichment for H3K4me3 and depleted H3K27me3. However, active promoters are marked by H3K4me3, and it could be a coincidence that NEAT1 also binds some of these promoters. At minimum, they should examine if H3K4me3 is reduced or transcription is reduced upon NEAT1 depletion.

Response

We analyzed H3K4me3 epigenetic mark in the promoter of the NEAT1 bound genes DLX5, KLK5 and DSC2 whose expression is reduced upon NEAT1 depletion (Figure 3). As shown in new Figure 4G, NEAT1 depletion impairs H3K4me3 mark on those promoters. As additional proof of the specificity of the link between NEAT1 trans genomic sites and transcriptional activation of

epidermal genes, we found that epidermal differentiation-related genes (e.g. cornification, epidermal development and lipid metabolism genes) are enriched in top 25% of NEAT1 trans genomic sites (see new Figure 4E). These evidence, together with the evidence that important epidermal transcription factor (KLF4) and epigenetic factor (BRG1) are able to interact with NEAT1 (see new Figure 4H and point 4, referee #2) support the specificity of our model.

6) Reviewer: *“Is there any known interactions between NEAT1 and transcription factors or transcription machinery, which can support the activation role of NEAT1 for transcription?”*

Response

Please see point 4, referee #2.

Minor points:

i) Reviewer: *“In Fig 1H, they showed that histone H3 acetylation is increased on the NEAT1 locus upon p63 KD. They need to specify which acetylation marks were tested or specifically testing the ones that are known to associated with gene activation.”*

Response

In Figure 1H, we utilized an anti-Histone H3 antibody (Abcam #ab47915) which specifically recognize histone H3 when is acetylated in K9, K14, K18, K23 or K27 residues. All these epigenetic marks have been associated with transcriptional activation (Grunstein M. et al., 1997). By analyzing publicly available ChIP-seq data we found that histone H3K9ac and Histone H3K27ac epigenetic marks are localized in the p63 binding sites of *NEAT1* and *MALAT1* loci and H3K27ac signal intensity increases in differentiated keratinocytes in concomitance with the decreased expression of Δ Np63 (see new Figure S2F).

ii) Reviewer: *“In p63 KD experiment, is HDAC1 binding to MALAT1 and NEAT1 loci reduced? They should measure this and demonstrate the correlation between reduced HDAC1 binding and increased NEAT1 expression”*

Response

In new Figure S2E we demonstrated that p63 silencing decreases the binding of HDAC1 on NEAT1 locus. This effect is parallel to the increase of NEAT1 RNA levels upon p63 silencing (see Figure 1C and 1D).

iii) Reviewer: *“In Fig. 4, they used ChIRP-seq to identify NEAT1 bound genomic regions. They should provide the mapping details for how many regions are mapped and show more detailed mapping results in supplemental data. More importantly, they should analyze whether there are any consensus motifs in those NEAT1 bound regions.”*

Response.

We added the mapping details of the ChIRP-seq analysis (see new Figure S7A). Regarding the consensus motif in the NEAT1 trans genomic region, see referee #1, point 4.

iv) Reviewer: *“Does NEAT1 bind to these regions through RNA:DNA interaction or through additional RNP?”*

Response.

We thank the reviewer to raise this intriguing question. As shown in the new Figure S8, NEAT1 trans genomic sites are enriched for the DNA binding motifs of transcription factors, such as ZKSCNA3 which has been reported to be associated with the essential paraspeckles component ZNF24 (REF). Moreover, we found that in differentiated keratinocytes NEAT1 binds to critical

players of epidermal differentiation such the epigenetic factor BRG1 or the epidermal transcription factor KLF4 (see new Figure 4H). Therefore, it is likely that NEAT1 binding to DNA occurs through associated proteins. However, we can rule out the possibility of a direct association of NEAT1 with DNA since a previous report demonstrated the ability of NEAT1 to bind the DNA in the absence of proteins (Senturk Cetin et al., 2019). Along NEAT1 sequence there are indeed significant triplex-forming regions (TFR) with the ability to interact with specific DNA sequence (Senturk Cetin et al., 2019). However, this study has been performed *in vitro* and it is not clear whether a similar scenario occurs *in vivo*. We discussed this point (see Discussion, page 14).

v) Reviewer: *“They should provide a global view of how many gene promoters are bound by NEAT1 and how their expression is changed upon NEAT1 KD. Are most NEAT1 bound genes downregulated or only a few?”*

Response.

By RNA-seq analysis we identified a discrete number (219) of genes whose expression is downregulated by NEAT1 depletion. NEAT1 trans genomic sites are present all over the genome and NEAT1 binds approximately 6000 gene promoters if we consider all NEAT binding sites, and 2000 gene promoters considering the top 25% only. We found that 41 genes are downregulated by NEAT1 and bound by NEAT1. GO term analyses revealed that these 40 genes categorize in GO terms associated with epidermis development, establishment of skin barrier and skin development (see new Figure S7B). Based on these data, we can argue that NEAT1 binding to the promoter regions does not imply per se transcriptional activation and it is likely that the NEAT1 effect on specific subset of gene promoters is dictated by its interaction with epidermal factors (for instance KLF4) (see also previous point). By analyzing the intersection between NEAT1 TSS bound genes and the RNA profile of the keratinocytes differentiation we found that that NEAT1 is preferentially found on the TSS of epidermal genes transcriptionally induced respect to those down regulated (651 vs 116) upon differentiation (see new Figure 4D). Furthermore, we found that epidermal differentiation-related genes (*e.g.* cornification, epidermal development and lipid metabolism genes), whose expression is downregulated upon NEAT1 depletion, are enriched in top 25% of NEAT1 trans genomic sites (see new Figure 4E), meaning that NEAT1 binding to specific subset of gene promoters is required for efficient transcription. This specificity may be dictated by NEAT1 binding to epidermal factors which facilitate the expression of specific genes (see also previous point). Accordingly, out of 219 genes downregulated by NEAT1 depletion, the subgroup of NEAT1 bound genes (40) are critical player of epidermal differentiation program (see new Figure S7B).

vi) Reviewer: *“In the 3D culture (Fig. 5C), the basal layers appear to be abnormal. More careful studies should be done with additional basal markers such as Ecad, Krt5, basement membrane, proliferation and apoptosis markers etc”*.

Response

We analysed integrin $\alpha6\beta4$ and laminin localization, two basal epithelia markers (see new Figure 5F and S11). We did not observe any alterations of the expression and localization of these additional basal markers. Regarding proliferation and apoptosis markers, please see previous point 3 and new Figure S9.

We would like to thank you this referee for the constructive criticisms He/She raised. We hope that this revised version may be satisfactory for this referee.

References

Bao X, Tang J, Lopez-Pajares V, Tao S, Qu K, Crabtree GR, Khavari PA. ACTL6a enforces the epidermal progenitor state by suppressing SWI/SNF-dependent induction of KLF4. *Cell Stem Cell*. 2013 Feb 7;12(2):193-203.

Boxer LD, Barajas B, Tao S, Zhang J, Khavari PA. ZNF750 interacts with KLF4 and RCOR1, KDM1A, and CTBP1/2 chromatin regulators to repress epidermal progenitor genes and induce differentiation genes. *Genes Dev*. 2014 Sep 15;28(18):2013-26

Bruno T, Corleone G, Catena V, Cortile C, De Nicola F, Fabretti F, Gumenyuk S, Pisani F, Mengarelli A, Passananti C, Fanciulli M. AATF/Che-1 localizes to paraspeckles and suppresses R-loops accumulation and interferon activation in Multiple Myeloma. *EMBO J*. 2022 Nov 17;41(22):e109711.

C. M. Clemson, J. N. Hutchinson, S. A. Sara, A. W. Ensminger, A. H. Fox, A. Chess and J. B. Lawrence: An architectural role for a nuclear noncoding RNA: NEAT1 RNA is essential for the structure of paraspeckles. *Mol Cell*, 33(6), 717-26 (2009)

Chakravarty D, Sboner A, Nair SS, Giannopoulou E, Li R, Hennig S, Mosquera JM, Pauwels J, Park K, Kossai M *et al* (2014) The oestrogen receptor alpha-regulated lncRNA NEAT1 is a critical modulator of prostate cancer. *Nat Commun* 5: 5383

Chu C, Quinn J, Chang HY. Chromatin isolation by RNA purification (ChIRP) Journal of visualized experiments : JoVE 2012

Chujo T, Yamazaki T, Kawaguchi T, Kurosaka S, Takumi T, Nakagawa S, Hirose T. Unusual semi-extractability as a hallmark of nuclear body-associated architectural noncoding RNAs. *EMBO J*. 2017 May 15; 36(10): 1447–1462.

Clark MB, Johnston RL, Inostroza-Ponta M, Fox AH, Fortini E, Moscato P, Dinger ME, Mattick JS. Genome-wide analysis of long noncoding RNA stability. *Genome Res*. 2012 May;22(5):885-98.

Fong KW, Li Y, Wang W, Ma W, Li K, Qi RZ, Liu D, Songyang Z, Chen J. Whole-genome screening identifies proteins localized to distinct nuclear bodies. *J Cell Biol* 2013 Oct 14;203(1):149-64.

Grunstein M. Histone acetylation in chromatin structure and transcription. *Nature*. 1997;389:349–352.

Gupta S, Stamatoyannopoulos JA, Bailey TL & Noble. *WS Genome Biology* volume 8, Article number: R24 (2007)

Indra AK, Dupé V, Bornert JM, Messaddeq N, Yaniv M, Mark M, Chambon P, Metzger D. Temporally controlled targeted somatic mutagenesis in embryonic surface ectoderm and fetal epidermal keratinocytes unveils two distinct developmental functions of BRG1 in limb morphogenesis and skin barrier formation. *Development*. 2005 Oct;132(20):4533-44.

Kawaguchi T, Tanigawa A, Naganuma T, Ohkawa Y, Souquere S, Pierron G, Hirose T. SWI/SNF chromatin-remodeling complexes function in noncoding RNA-dependent assembly of nuclear bodies. *Proc Natl Acad Sci U S A*. 2015 Apr 7;112(14):4304-9.

LeBoeuf M, Terrell A, Trivedi S, Sinha S, Epstein JA, Olson EN, Morrisey EE, Millar SE (2010) Hdac1 and Hdac2 act redundantly to control p63 and p53 functions in epidermal progenitor cells. *Developmental cell* 19: 807-818

Lin-Shiao, E., Lan, Y., Coradin, M., Anderson, A., Donahue, G., Simpson, C.L., Sen, P., Saffie, R., Busino, L., Garcia, B.A., Berger, S.L., Capell, B.C., 2018. KMT2D regulates p63 target enhancers to coordinate epithelial homeostasis. *Genes & development* 32, 181-193.

Mardaryev AN, Gdula MR, Yarker JL, Emelianov VU, Poterlowicz K, Sharov AA, Sharova TY, Scarpa JA, Joffe B, Solovei I, Chambon P, Botchkarev VA, Fessing MY. p63 and Brg1 control developmentally regulated higher-order chromatin remodelling at the epidermal differentiation complex locus in epidermal progenitor cells. *Development*. 2014 Jan;141(1):101-11. doi: 10.1242/dev.103200.

Mello SS, Attardi LD (2018) Neat-en-ing up our understanding of p53 pathways in tumor suppression. *Cell Cycle* 17: 1527-1535

Mello SS, Sinow C, Raj N, Mazur PK, Biegging-Rolett K, Broz DK, Imam JFC, Vogel H, Wood LD, Sage J *et al* (2017) Neat1 is a p53-inducible lincRNA essential for transformation suppression. *Genes & development* 31: 1095-1108

Mercer TR, Qureshi IA, Gokhan S, Dinger ME, Li G, Mattick JS, Mehler MF (2010) Long noncoding RNAs in neuronal-glia fate specification and oligodendrocyte lineage maturation. *BMC Neurosci* 11: 14

Naveed A, Cooper JA, Li R, Hubbard A, Chen J, Liu T, Wilton SD, Fletcher S, Fox AH. NEAT1 polyA-modulating antisense oligonucleotides reveal opposing functions for both long non-coding RNA isoforms in neuroblastoma. *Cell Mol Life Sci*. 2021 Mar;78(5):2213-2230.

Panatta E, Lena AM, Mancini M, Smirnov A, Marini A, Delli Ponti R, Botta-Orfila T, Tartaglia GG, Mauriello A, Zhang X, Calin GA, Melino G, Candi E. Long non-coding RNA uc.291 controls epithelial differentiation by interfering with the ACTL6A/BAF complex. *EMBO Rep*. 2020 Mar 4;21(3):e46734.

Ramsey, M.R., He, L., Forster, N., Ory, B., Ellisen, L.W., 2011. Physical association of HDAC1 and HDAC2 with p63 mediates transcriptional repression and tumor maintenance in squamous cell carcinoma. *Cancer research* 71, 4373-4379.

S. Souquere, G. Beauclair, F. Harper, A. Fox and G. Pierron: Highly ordered spatial organization of the structural long noncoding NEAT1 RNAs within paraspeckle nuclear bodies. *Mol Biol Cell*, 21(22), 4020-7 (2010)

Sunwoo H, Dinger ME, Wilusz JE, Amaral PP, Mattick JS, Spector DL (2009) MEN epsilon/beta nuclear-retained non-coding RNAs are up-regulated upon muscle differentiation and are essential components of paraspeckles. *Genome Res* 19: 347-359

T. Naganuma, S. Nakagawa, A. Tanigawa, Y. F. Sasaki, N. Goshima and T. Hirose: Alternative 3'-end processing of long noncoding RNA initiates construction of nuclear paraspeckles. *EMBO J*, 31(20), 4020-34 (2012)

Tomohiro Yamazaki, Tetsuro Hirose. The building process of the functional paraspeckle with long non-coding RNAs. *Frontiers in Bioscience, Elite*, 7, 1-47, January 1, 2015.

Wen S, Wei Y, Zen C , Xiong W, Niu Y, Zhao Y. Long non-coding RNA NEAT1 promotes bone metastasis of prostate cancer through N6-methyladenosine. *Mol Cancer* 2020 Dec 12;19(1):171.

Zeng C, Xu Y, Xu L, Yu X, Cheng J, Yang L, Chen S, Li Y (2014) Inhibition of long non-coding RNA NEAT1 impairs myeloid differentiation in acute promyelocytic leukemia cells. *BMC Cancer* 14: 693

REVIEWERS' COMMENTS

Reviewer #1 (Remarks to the Author):

I applaud (and thank) the authors for their huge commitment to address the concerns so thoroughly and impressively. The authors really have gone above and beyond to address all my comments on the original manuscript in full.

As far as I am concerned, there is nothing left to ask. The conclusions are strongly supported in the revised manuscript that, in my opinion, meets all the criteria to be accepted for publication.

Reviewer #2 (Remarks to the Author):

The authors have adequately addressed my initial comments. I approve the publication of the manuscript in Nature Communications without additional changes.

Reviewer #3 (Remarks to the Author):

The authors have reasonably addressed my concerns. Although it is unfortunate that plasmid based reporter assays were unable to recapitulate the inhibitory effect of p63 on the promoter of NEAT1 and MALAT1, they provided additional evidence to support p63-dependent recruitment of HDAC1 to NEAT1 promoter. I therefore have no more concerns for this study.